# Macroalgal Proteins: A Review

**DOI:** 10.3390/foods11040571

**Published:** 2022-02-16

**Authors:** Ronan O’ Brien, Maria Hayes, Gary Sheldrake, Brijesh Tiwari, Pamela Walsh

**Affiliations:** 1Department of Food BioSciences, Teagasc Food Research Centre, Ashtown, D15 DY05 Dublin, Ireland; ronan.obrien@teagasc.ie; 2School of Chemistry and Chemical Engineering, Queen’s University Belfast, Belfast BT9 5AG, UK; g.sheldrake@qub.ac.uk; 3Department of Food chemistry and Technology, Teagasc Food Research Centre, Ashtown, D15 DY05 Dublin, Ireland; brijesh.tiwari@teagasc.ie; 4School of Mechanical and Aerospace Engineering, Belfast BT9 5AJ, UK; pamela.walsh@qub.ac.uk

**Keywords:** seaweed, peptides, techno-functional ingredients, health, extraction methods, digestibility, bioavailability, ACE-1 inhibition, dipeptidyl peptidase IV, cyclooxygenase enzymes

## Abstract

Population growth is the driving change in the search for new, alternative sources of protein. Macroalgae (otherwise known as seaweeds) do not compete with other food sources for space and resources as they can be sustainably cultivated without the need for arable land. Macroalgae are significantly rich in protein and amino acid content compared to other plant-derived proteins. Herein, physical and chemical protein extraction methods as well as novel techniques including enzyme hydrolysis, microwave-assisted extraction and ultrasound sonication are discussed as strategies for protein extraction with this resource. The generation of high-value, economically important ingredients such as bioactive peptides is explored as well as the application of macroalgal proteins in human foods and animal feed. These bioactive peptides that have been shown to inhibit enzymes such as renin, angiotensin-I-converting enzyme (ACE-1), cyclooxygenases (COX), α-amylase and α-glucosidase associated with hypertensive, diabetic, and inflammation-related activities are explored. This paper discusses the significant uses of seaweeds, which range from utilising their anthelmintic and anti-methane properties in feed additives, to food techno-functional ingredients in the formulation of human foods such as ice creams, to utilising their health beneficial ingredients to reduce high blood pressure and prevent inflammation. This information was collated following a review of 206 publications on the use of seaweeds as foods and feeds and processing methods to extract seaweed proteins.

## 1. Introduction

According to the Food and Agricultural Organisation (FAO), the projected supply of protein required by Europe in 2054 will be approximately 56 million metric tonnes [1]. To meet this demand, more alternative protein sources are required. Seaweed-derived protein may offer a viable solution as seaweeds may be considered sustainable, highly nutritious and they do not require land for cultivation [2]. Seaweeds have a reported protein content of between 8 and 47% dry weight, and the highest protein content is found in red and green seaweed species [3]. The protein content in some seaweed has been reported to be greater than traditional protein sources, including milk (3.4%) [4,5]. However, seaweeds can also contain bioactive compounds that may have antinutritional effects if consumed in high concentrations (usually above 4% dry matter intake for animal feeds) including phlorotannins, polyphenols and lectins as well as saponins and flavanoids. These also have bioactivities and health effects associated with their consumption in suitable concentrations.

Seaweed can grow in environments that are not suitable for other plants/vegetation such as high-pressure zones and in the presence of high salt concentrations [6]. In addition to their value as a potential protein source, seaweeds are also rich in phytochemicals (e.g., terpenoids and phlorotannins) [7], and a sustainable resource for essential vitamins and amino acids [8]. They have the potential to be considered “climate-friendly” sources of protein that may help to achieve the UN sustainability goal of zero hunger by 2030 [9].

Seaweeds can be classified into three groups: Rhodophyta or red seaweeds, Phaeophyta or brown seaweeds and Chlorophyta or green seaweeds. This classification is based on their unique pigments, which are chlorophylls (green seaweeds), fucoxanthin (brown seaweeds) and phycobilin (red seaweeds) [10]. The protein content of seaweed varies and is dependent on the species, location, and time of harvest [11] in addition to variations due to extraction and characterisation methods. Red seaweeds, such as *Palmaria palmata*, have reported protein contents between 8 and 47% of the dry weight of the seaweed [3,12]. Green seaweeds have a reported protein content ranging from 9 to 26% of the dry weight of the seaweed [13,14]. The protein content of brown seaweed is less than red or green and is usually between 3 and 15% of the dry weight of the seaweed [11].

In addition, seaweeds have attracted significant research interests due to their observed bioactivities and potential health benefits. This is due, in part, to the presence of natural bioactive peptides in seaweeds [15]. Bioactive peptides are small protein fragments made up of between 2 and 20 amino acids which can play an important role in regulatory processes in humans including the immune, cardiovascular, and nervous systems [16]. These bioactive peptides may result in bioactivities including anti-diabetic, antioxidant, anti-microbial, and anti-hypertensive activities [17]. Bioactive peptides are inactive before cleavage from the parent protein but can be generated using proteolytic enzymes such as pepsin (EC 3.4.23.1), trypsin (EC 3.4.21.4), chymotrypsin (EC 3.4.21.1), and other methods including high-pressure or lactic acid fermentation of whole-seaweed biomass or extracted seaweed proteins [17,18,19,20,21].

Seaweeds are a source of bioactive peptides that can effect enzymes that play an important role in the control of diabetes mellitus and blood pressure. Enzymes include angiotensin-I-converting enzyme (ACE-1; EC 3.4.15.1) and renin (EC 3.4.23.15), which operate within the renin–angiotensin–aldosterone (RAAS) system [22,23,24] and control high blood pressure and salt-water balance. In Japan, China, and Korea, seaweed is consumed as part of a healthy diet and lifestyle. It is noteworthy that these populations have reduced incidences of cardiovascular disease and diabetes mellitus compared to Western societies [25,26].

The release of bioactive peptides from seaweed protein is dependent upon the breakdown of polysaccharides in the cell wall, which can vary depending on the type of seaweed. The primary polysaccharide found in red seaweeds is carrageenan, whereas fucoidan is found in most brown seaweeds, and ulvans in green seaweeds [27]. A number of techniques have been used to disrupt the polysaccharide cell wall. Different methodologies to break the cell wall include autoclave treatment, ultrasound/sonication, high-pressure processing, and pulsed electric field extraction-based approaches [10]. However, most methods used to date have failed to isolate proteins and bioactive peptides from seaweeds in economically viable yields [28,29]. Enzyme-assisted extraction of protein and bioactive peptides has amassed interest due to high specificity and potential preservation of target compounds, along with minimal use of solvents, reduced energy consumption, and improvement in extractable protein yield and the upscale potential of this approach [10,30,31,32]. Enzyme-assisted extraction employs the use of hydrolytic enzymes that disrupt the cell wall polysaccharide structure and storage compounds, leading to the release of intercellular proteins and peptides [33]. A number of recent publications have reported improvements in extractible protein yields and the generation of bioactive peptides using enzyme hydrolysis [34,35,36].

In the food industry, seaweeds are consumed whole and dried, in pasta and bread products, as salads and as supplements. Extracts from seaweeds are used as weight control agents and as antioxidants and are used to make agar for microbiology purposes as well as food ingredients due to their techno-functional attributes. Macroalgal proteins are used in the manufacture of alternatives to meat patties and burgers and macroalgal proteins also find use in egg substitute products including mayonnaise and eggs.

The novelty of this review is that it discusses macroalgal proteins for use in foods and feeds. This review collates the existing detailed information from previously published literature on methods for extraction of proteins and bioactive peptides from red, green and brown seaweeds using enzymes and potential applications of extracted proteins and peptides in an objective review of over 300 papers with original papers cited. Depending on the extraction approach, different yields and types of proteins/peptide hydrolysates are obtained. This results in varying functionalities and bioactivities are associated with the proteins/peptides generated. In addition, co-products of seaweed protein extraction including tannins, phenols and other bioactives (depending on the species used) that could have potential for use in the pharmaceutical and cosmeceutical industries and applications of extracts generated will be discussed.

### 1.1. Protein Extraction Procedures Applied to Seaweeds

The extraction of protein and the isolation of bioactive peptides are particularly challenging as although several extraction procedures have been tested in recent years, there have been issues surrounding the generation of protein in economically viable yields [37]. Therefore, there has been a growing trend of novel techniques that have been designed to improve the yields of protein obtainable from seaweed biomass [38]. Ideally, these methods should be quantitative, cost effective, non-destructive and not time consuming [7]. Table 1 highlights the advantages and disadvantages of several trialled protein extraction procedures applied to seaweeds to date.

#### 1.1.1. Search Methodology

A number of search engines such as Google Scholar, PubMed and Web of Science were utilised to explore the established algal-based literature, shortlisting significant findings, which were represented in tabular data.

#### 1.1.2. Enzyme Hydrolysis

Enzyme-assisted extraction utilises enzyme hydrolytic action in order to disrupt the cell wall structure, causing the breakdown of storage components and releasing embedded proteins/peptides as shown in Figure 1 [51,52]. Enzyme-assisted extraction methods are reported to increase protein yields as well as preserve bioactive components and metabolites [29,53]. Enzyme-assisted extraction disrupts the cell wall by the digestion of specific polymer bonds and commonly uses enzymes such as cellulases (EC 3.2.1.4), xylanases, and alcalases^®^ (EC 3.4.21.14) (shown in Table 2) that disrupt polysaccharide components of the algal cell wall either alone, or as part of a cocktail mixture [54,55,56]. There are several variable factors that affect the activity of enzymes including the time–temperature interval [57], the optimum temperature and pH, and the amount of agitation applied during hydrolysis [7]. Enzyme-assisted extraction has considerable advantages compared to other techniques. For example, carbohydrase and protease enzymes have high catalytic efficiency, improve the overall extraction yields of proteins, use minimal levels of organic solvents, and make the extraction of protein low cost [56].

#### 1.1.3. Microwave-Assisted Extraction

Microwave-assisted extraction utilises non-ionising electromagnetic radiation in order to enhance the extraction of protein from seaweed biomass through the vibration of water molecules in the extraction solution [27]. This results in the disruption of hydrogen bonds and the migration of dissolved ions, resulting in the extraction of selectively targeted compounds [39,70]. This low-energy, efficient technique has been employed quite successfully in the extraction of carbohydrate or phenolic compounds from brown seaweeds including *Ascophyllum nodosum* and *Laminaria japonica* [13,39,71]. Its limiting factor is the moisture content of dried samples, whereby the recoverable yield can be lower than expected [72] and, moreover, thermo sensitive compounds such as polysaccharides, peptides and proteins may be destroyed following extraction [39,43]. Potentially optimising specific parameters such as the duration and temperature during microwave-assisted extraction may have a positive effect on the overall protein extraction yield as shorter processing times (5 min) at 187 °C may be the key to successful protein extraction from seaweed biomass as demonstrated by Magnusson et al. with *Ulva ohnoi* [58].

#### 1.1.4. Ultrasound Sonication Extraction

Ultrasound sonication techniques have been used to extract protein from seaweed biomass through the use of acoustic cavitation generated from soundwaves, which creates microturbulence in the form of air bubbles. This results in the generation of mechanical energy in the form of shear forces that can be harnessed to break down macroalgal cell wall polymers while enhancing the release of target molecules [73,74,75,76]. Ultrasound sonication is an environmentally friendly technique using minimal amounts of organic solvents; additionally, it is a fast technique that requires low energy [44]. There are two types of ultrasound equipment that can be used including inexpensive sonication water baths and ultrasonic probes that have increased power [10]. It has been reported that depending on the operating power and specific parameters of sonication, some protein and polysaccharide structures can be degraded as a result of ultrasound extraction [10,77].

#### 1.1.5. Pulse Electric Field Extraction

Pulsed electric field-based extraction uses high-intensity voltage (kV) pulses to alter the cell membrane of seaweed biomass, resulting in the reversible/irreversible membrane permeation of intracellular matter such as proteins and peptides from seaweed biomass as a result of electroporation [78,79,80]. This process has several advantages including that it is energy efficient, environmentally friendly, and can be operated at low temperatures, meaning undesirable change to potentially heat-sensitive target components [47,81,82] such as peptides and proteins can be avoided, resulting in easier downstream purification following extraction [80]. A pulsed electric field can be operated under two modes, batch and continuous, and has potential to be scaled up [83]. However, factors such as conductivity needed for extracted medium can be high in some cases, which may limit use [46].

#### 1.1.6. Solid–Liquid Extraction

Solid–liquid extraction is the traditional method of protein-based extractions from seaweeds using deionised water and solvents as shown in Figure 2 [30]. The use of aqueous medium of different acidic and alkaline solubility conditions can be used to promote the overall protein yield from seaweed. Certain factors such as time, temperature, and weight to volume ratio of seaweed biomass to solvent [28] have to be optimised. The shift to pH-based protein extraction has been employed in a number of methods, leading to increases in overall protein extraction yields [49]. These methods are quite simple and cost effective to operate [50]. Limiting factors include higher temperatures, as only low temperatures during extraction ensure undesirable damage to proteins but, the process can also be time consuming [48].

### 1.2. Green Seaweed

#### 1.2.1. Protein Content and Seasonality

Green seaweeds have protein contents of between 9 and 26% of the dry mass of the seaweed depending on factors such as location of harvest and time of season [13]. The green seaweeds *Ulva lactuca* and *Ulva rigida* were previously reported to contain 32.7% and 32% protein (dry mass basis), with the highest protein values for seaweeds harvested in August, while lower values were observed in seaweeds harvested in April (location France) [11]. In general, green seaweeds contain essential amino acids, aspartic acid, glutamic acids, alanine, histidine, arginine, leucine, and threonine [84]. *Ulva lactuca*, also known as “sea lettuce”, and *Caulerpa* are rich in aspartic and glutamic acids as well as alanine, histidine, and arginine [85]. *Codium fragile* is particularly rich in valine, glutamic acid, and methionine amino acids [86]. Additionally, *Cladophora rupestris* contains the conditional amino acid, taurine [87]. Taurine plays an active role in the excretion of bile, resulting in a reduction in cholesterol due to the absorption of lipids [88,89,90]. The protein quality of these seaweeds can be potentially limited by polysaccharide fibres to which proteins are bound, thus preventing the action of proteolytic enzymes and the release of peptides from the parent protein [11].

#### 1.2.2. Documented Protein Extraction Methods Applied to Green Seaweeds

The main problem encountered extracting protein and bioactive peptides from green seaweed is the failure to breakdown the anionic cell wall composed of various polysaccharide components including starch, ulvans, galactans, xylans, and mannans, as well as polyphenolic compounds [91]. Proteins are bound to these polysaccharide components via hydrogen bonding, and therefore disruption to the seaweed cell wall is imperative to significantly improve protein yield [13,31,55]. A number of physical, chemical, and enzymatic methods were applied previously to green seaweeds as outlined in Table 1. 

##### Physical Extraction Methods Applied to Green Seaweeds

Several physical extraction methods that have been used previously utilised pressure, ionising radiation, and heat in order to extract proteins from green seaweeds shown in Table 2. One method used microwave-generated radiation to weaken the bonds within the polysaccharide cell wall. Magnusson et al. carried out microwave-assisted extraction of proteins from *Ulva ohnoi* and reported a protein yield of 23.9% (dry weight) [58]. Additionally, ultrasound waves were used to extract protein from green seaweeds as the generation of cavitation bubbles from ultrasound waves elicits particle breakdown of the cell wall within seaweed biomass and simulates the release of bioactive compounds [39,92,93]. Rahimi et al. reported a yield of 11.2% proteins from the macroalgae *Ulva intestinalis* using this method [93]. Ultrasound-assisted extraction has been combined with enzyme-assisted extraction. Rodrigues et al. extracted proteins from *Codium tomentosum* and reported a yield of 16.9−18.8% protein following hydrolysis using several enzymes including endoprotease, exopeptidase, and flavourzyme^®^ combined with ultra-sonication [37].

##### Chemical Extraction Methods Applied to Green Seaweeds

Chemical extraction methods include solvent extraction in acidic or alkaline conditions. These were used previously to precipitate protein from specific seaweeds as outlined in Table 2. Angell and colleagues utilised different ratios of aqueous solvent in alkaline conditions in order to extract protein from *Ulva ohnoi*. Seaweed biomass was exposed to different ratios of NaCl (20:1 or 5:1 *v*/*w*) at pH 12, incubated for different time periods and temperatures and subsequentially centrifuged. The precipitated protein pellet reported a protein yield of between 12.28 and 21.57% [59]. Additionally, some methods have utilised a number of pH ranges from acidic (pH 2) to basic (pH 7), and alkaline (pH 12) (method, also known as the pH shift method) to extract proteins from seaweed biomass. Harrysson et al. employed the use of the pH shift method to extract protein from the green seaweed *Ulva lactuca*.

The pH of the seaweed biomass was adjusted using NaOH and HCl. Acidic conditions produced the best results in terms of protein yield and content. A protein yield of 22.7% dry mass of the seaweed was reported by Harrysson et al. [49].

##### Enzyme-Assisted Extraction Methods Applied to Green Seaweeds

Enzyme-assisted extraction methods apply specific polysaccharide-degrading enzymes to the seaweed cell walls in combination with protease enzymes to release proteins/small proteins and peptides from the biomass to the supernatant, which is collected following centrifugation [35]. Hydrolysis conditions of temperatures, pH, and mixing are controlled [94]. This approach is more specific compared to conventional extraction methods and was reported to result in increased protein and bioactive peptide yield. Enzyme-assisted extraction is considered a cleaner method as minimal use of solvent is required [30]. Previously, Hardouin et al. utilised a number of hydrolytic enzymes including endoproteases, cellulases (EC 3.2.1.4), xylanases (EC 3.2.1.8), β-glucanases (EC 3.2.1.6), and arabanases (*EC* 3.2.1.99) on the green seaweed species *Ulva* sp. A protein yield of 24.4% dry weight was recorded [34]. Moreover, Hardouin and colleagues repeated this enzyme-assisted extraction technique with another *Ulva* species, reporting a range of protein yields from *Ulva armoricana* of between 6.2 and 10.1% [35]. Garcia-Vaquero and colleagues recorded a protein yield of 69.1% based on the dry weight of the seaweed biomass from *Ulva latuca* using the food-grade enzyme papain (EC 3.4. 22.2) [6]. This indicates species specific protein yield differs within *Ulva* seaweeds.

##### Bioactivities of Generated Proteins/Peptides Derived from Green Seaweeds

Bioactivities reported for green seaweed proteins and peptides include anti-inflammatory and anti-hypertensive activities as shown in Table 3. Several peptides and proteins generated from *Ulva* have reported angiotensin-I-converting enzyme (ACE-1) as well as renin (EC 3.4.23.15) inhibitory activities including peptides from *Ulva lathrata, Ulva intestinalis, Ulva lactuca,* and *Ulva rigida* [95,96,97]. Renin and ACE-1 are the two key regulatory enzymes within the renin–angiotensin–aldosterone system (RAAS), which is used to regulate blood pressure of mammals [98]. The generation of renin results in the increased production of angiotensin I and ACE-1 is responsible for the conversion of angiotensin I to the potent vasoconstrictor angiotensin II, which results in elevated blood pressure and hypertension activity [99,100,101]. Sun et al. reported a number of anti-hypertensive peptides including FGMPLDR and MELVLR derived from *Ulva intestinalis* hydrolysates with trypsin (EC 3.4.21.4), pepsin (EC 3.4.23.1) and papain enzymes [95]. Additionally, anti-hypertensive peptides were reported from *U. lathrata* (the peptide PAFG) [96] and *U. rigida* (IP, AFL peptide sequence) [97]. Moreover, Garcia-Vaquero et al. identified 48 novel peptides from *U. lactuca* hydrolysates using papain, and RP-HPLC renin inhibitory peptides were also found [6].

Kazir et al. identified antioxidant peptides from *Ulva* sp.-derived peptides, where the oxygen radical absorbance capacity (ORAC) and ferric-reducing antioxidant power (FRAP) values were determined at 20 µM trolox equivalents/g protein, respectfully [102]. Anti-inflammatory activity has also been reported for *Ulva* sp. hydrolysates generated with the enzymes alkaline protease, protex 6L and flavourzyme^®^ (EC 232-752-2). Additionally, *Ulva* sp. was found to have anti-inflammatory effects when an *Ulva* sp. hydrolysate was generated using alkaline protease, protex 6L and flavourzyme^®^ enzymes [102]. *Ulva* sp. hydrolysates were also reported to increase cytokine IL-10 in rats [102].

##### Application of Green Seaweeds in Foods and Feeds

Seaweeds and seaweed-derived bioactives are typically used in the food industry as gelling and thickening agents as well as protein replacements and fish feeds as shown in Table 4. The consumption of *Ulva pertusa* by red sea bream (*Pagrus major*) was reported to increase fish growth and protect against parasitic disease in this fish species due to the immune activity of lipids found in this seaweed [11]. There have been a number of green seaweeds that have been utilised as supplementary food ingredients including *Caulerpa racemose*, *Ulva intestinalis*, *Ulva lactuca*, and *Ulva rigida*, which have been used to enhance the antioxidant properties within cereal-based products such as semi-sweet biscuits, bread, as well as pork patties, and fish surimi [8].

### 1.3. Red Seaweed

#### 1.3.1. Protein Content and Seasonality

Seasonal changes in protein content of different seaweeds are thought to occur because of factors such as water temperature, wave force and light intensity at the location and time of harvest [3]. *Palmaria palmata*, also known as Dulse, is one of the most studied red seaweed species, with a protein content that can vary between 8 and 35% of its dry weight and with the highest reported protein content from *Palmaria palmata* collected from Iceland in the winter months [87,130]. Other red seaweeds have indicated higher seasonal protein content during the winter months including *Porphyra dioica* [131]. Red seaweed has a rich amino acid composition containing all the essential amino acids [132]. Galland-Irmouli and colleagues have documented that the amino acid composition of *Palmaria palmata* was not affected by factors such as seasonality [130].

#### 1.3.2. Documented Protein Extraction Methods Applied to Red Seaweeds

Red seaweeds are composed of polysaccharides, xylans and cellulose within their cell walls [62]. Extraction methods aim to break down the β (1 → 4)/(1 → 3) linkages and the hydrogen bonding in red seaweed cell walls in order to generate higher protein yields [29].

##### Physical Extraction Methods Applied to Red Seaweeds

Several physical extraction methodologies have been employed to extract protein from different types of red seaweed (Table 2). O’ Connor et al. utilised autoclave and sonication applied to the red seaweeds *Palmaria palmata* and *Chondrus crispus*. Protein yields of 21.5% from *Palmaria palmata* and 35.5% from *Chondrus crispus* were reported, respectively [28]. Homogenisation-based methods have been attempted; for example, Barbarino et al. reported a protein yield of 8.9% dry weight from *Porphyra acanthophora* using a Potter homogeniser. Following homogenisation, the protein was precipitated using trichchloroacetic acid [61].

##### Enzyme-Assisted Extraction Methods Applied to Red Seaweeds

Macroalgae contain a number of polysaccharide components that make up the matrix where seaweed protein remains embedded, and these components have become the primary focus of extraction methods in order to facilitate the release of protein through the use of hydrolysing enzymes [10]. A number of methods have been attempted to enhance the extractible protein yields through the specific combination of carbohydrase and proteolytic enzymes [12,56,62,63]. Naseri et al. utilised several enzymes including Celluclast (EC 232-734-4), Shearzyme^®^, alcalase^®^, and viscozyme^®^ to extract protein from *Palmaria palmata* [12]. This method reported a protein yield of between 35.5 and 41.6% based on the dry mass (alcalase^®^ and Celluclast were used) [12]. Similarly, Bjarnadóttir et al. reported a yield of 33.4% protein based on the dry mass from *P. palmata* utilising the enzymes umamizyme and xylanase [62]. Pimentel et al. noted a protein yield of 23% using enzymes prolyve^®^1000, and flavourzyme^®^ from *Porphyra tenera* [63]. Vásquez and colleagues reported a high protein yield of 36.1% from *Chondracanthus chamissoi* using α-amylase, cellulose, and pectinase [55]. Moreover, Kulshreshtha et al. reported a yield of 7.1% protein from *Chondrus crispus* when cellulose was applied [60].

##### Bioactive Activities of Generated Proteins/Peptides

Red seaweed peptides have reported antioxidant, anti-coagulant, anti-diabetic, and anti-hypertensive activities as shown in Table 3. Suetsuna identified a number of ACE-1 inhibitory peptide sequences derived from *Porphyra* sp. protein hydrolysates purified using ion-exchange and gel-filtration techniques. The peptides were IY, MKY, AKYSY, and LRY [104]. Moreover, Furuta et al. reported ACE-1 inhibition activity from a number of peptides derved from *Palmaria palmata* hydrolysate generated using thermolysin enzyme [107]. Additionally, Indumathi et al. reported the anti-coagulant activity of a peptide with an amino acid sequence (NMEKGSSSVVSSRM) found in a hydrolysate made from pepsin (EC 3.4.23.1) applied to *Porphyra yezoensis* [105].

Antioxidant bioactivity was reported for peptides derived from *Palmaria palmata* hydrolysates [106]. Harnedy et al. reported a number of peptide sequences from a *Palmaria palmata* hydrolysate made with corolase and pepsin enzymes that had antioxidant activity. In particular, one peptide (SDITRPGGQM) had the highest oxygen radical absorbance capacity (ORAC) and ferric-reducing antioxidant power (FRAP) values [106]. In addition, *Palmaria palmata* protein hydrolysates containing the bioactive peptide (ILAP) and a *Porphyra dioica* protein hydrolysates containing the bioactive peptide (WLVA) were found to inhibit dipeptidyl peptidase IV (DPP IV), one of the key regulatory enzymes involved in the secretion of insulin in diabetes mellitus [10]. *Palmaria palmata* is also a source of renin inhibitory peptides. The tridecapeptide peptide IRLIIVLMPILMA was previously generated from this seaweed and was shown to inhibit renin [110]. Fitzgerald and colleagues noted a 33 mm Hg drop in blood pressure when this peptide was tested in vivo for anti-hypertensive effects using spontaneously hypertensive rats [110].

*Palmaria palmata* hydrolysates were reported to have platelet-activating factor acetylhydrolase (PAF-AH; EC 3.1.1.47) inhibition activity [111]. PAF-AH is a key enzyme in the oxidation of low-density fatty acids and is an early indicator of atherosclerosis commonly associated with strokes [133,134]. Fitzgerald et al. identified a peptide with the single amino acid sequence NIGK from *Palmaria palmata* when this seaweed was hydrolysed with papain [111]. Red seaweeds such as *Porphyra* sp. have also been reported to inhibit α-amylase activity and this is one of the key enzymes in glycaemic control in diabetes mellitus. Admassu et al. reported several protein hydrolysates and peptides (GGSL and ELS) made using pepsin and viscozyme^®^ that were found to inhibit α-amylase [112]. Anti-pain activity was reported for *Porphyra yezoensis* hydrolysates (PPY1 peptide sequence) as Lee and colleagues documented cyclooxygenase (COX) enzyme 2 inhibitory activity [113]. Cyclooxygenase enzyme is commonly associated with the development of inflammatory processes and pain through eicosanoid molecules such as prostaglandin [113,135].

##### Application of Red Seaweeds in Foods and Feeds

Red seaweeds are particularly used in the food industry as gelling agents as they are good sources of agar, with dominant sources being from the genus *Gelidium, Gracilaria,* and *Pterocladisa* [136]. There have been a number of red seaweed species including *Kappaphycus* sp. and *Eucheuma* sp. that have been used in the production of carrageenan that is used to improve flavour and appearance as well as extending the shelf life of food products such as meat, ice cream, dairy, and jellies [137,138]. Red seaweeds have been incorportated into animal feeds as *Phyllophara* sp. has been utilised in dairy cows in order to improve the yields of milk, resulting in a further increase of 4.4% in overall milk yields and a rise in overall fat content by 0.24% [127]. Additionally, red seaweed species *Porphyra yezoensis* has been supplemented into the diets of red seabream, showing positive results in terms of an increase in overall growth when incorporated at 5% [128]. A number of functional foods containing bioactive peptides from red seaweeds are known and sold in Japan. Riken Vitamin Co and Shirako have established anti-hypertensive health claims as associated with the consumption of wakame peptide jelly and Nori peptide S. These health claims have been confirmed by the Japanese Ministry of Health and Welfare in Japan [139,140]. Red seaweed protein derived from red macroalgae including *Grateloupia turuturu* have the potential to be used in human food applications including R-phycoerythrin and R-phycocyanin that [128] can be used in food colouring applications. Moreover, Fitzgerald et al. successfully incorporated *palmaria palmata* protein hydrolysate into bread, resulting in no adverse changes in appearance, texture, moisture and colour of the bread [141].

Red seaweeds also have applications as feed ingredients for animals and fish/aquaculture. In recent times, they are used for their anti-methane effects when used as additives in the diet of ruminants including cattle, sheep and dairy cows. Several authors have reported that the red seaweed Asparagopsis taxiformis can reduce methane emissions by as much as 98% when added to feeds at concentrations between 0.5 and 4% of the total dry matter intake [142,143]. However, the active agent in this instance is the bioactive compound bromoform—a known carcinogen. Red seaweeds can also be used in animal feeds as a source of anthelmintic agents. Recently, the anthelmintic action of a traditional remedy developed in Italy consisting of the red seaweeds *Palisada tenerrima, Laurencia intricata* and *Laurencia* spp. red algae was assessed using the egg hatch test [144]. An egg hatch reduction of 89.5, 43.7, and 23.4% was observed at 50, 5 and 1% dilutions [144].

### 1.4. Brown Seaweed

#### 1.4.1. Protein Content and Seasonality

Brown seaweeds have protein contents of between 3 and 15% of their dry weight. However, there are some exceptions such as *Undaria* sp. and *Sargassum* sp. that were reported to contain up to 24% protein [145]. Brown seaweeds are reliable sources of essential amino acids, but it has been reported that there are little to non-existent levels of methionine, cysteine, and lysine amino acids found in these seaweeds [28]. It has been documented that protein content in brown seaweeds is generally higher in winter than in summer as the seaweed plant undergoes its developing phase, resulting in increased photosynthetic activity [6,90]. However, Kumar et al. indicated seasonal protein changes in *Sargassum wightii*, whereby protein yields recorded during winter and early spring months were higher than at any other time as 12.7% dry mass were recorded in March [65]. The differences in growth and protein yield of seaweeds can be due to the uptake of nutrients such as nitrogen that mainly occurs during the winter months [146,147].

#### 1.4.2. Documented Protein Extraction Methods Applied to Brown Seaweeds

In brown seaweed, the complex cell wall, comprising alginate, fucoidan, cellulose, laminarin and mannitol, acts as the barrier to efficient protein extraction as proteins are embedded within the polysaccharide matrix [28]. Additionally, phenolic compounds such as phlorotannins that are exclusively found in brown seaweeds can also impact protein extraction efficiency due to the complex bonding in which proteins are bound to phenolic compounds within the algal matrix [148]. Therefore, a number of methodologies have been applied to break this bonding within brown seaweeds as outlined in Table 2.

##### Physical Extraction Methods Applied to Brown Seaweeds

Several physical extraction methods have been utilised to extract protein from brown seaweeds including autoclave, osmotic shock, and high-pressure processing [28]. O’Connor et al. employed a number of physical pre-treatments including high-pressure processing and autoclave-based techniques that resulted in protein yields of 23.7% and 24.3% based on dry weight in *Fucus vesiculosus*, respectively. Moreover, high-pressure processing and autoclave-based protein extraction were applied to *Alaria esculenta*, with reported protein yields of 15% and 17.1% dry weight of seaweed biomass [28]. Ultrasound-assisted extraction has also been used in conjunction with traditional chemical extraction, as Kadam et al. reported a protein yield of 7.1% from *Ascophyllum nodosum* [66]. Ultrasound methodologies were used in conjunction with acid and alkaline extraction. Alkali treatment with NaOH resulted in higher protein recovery [66].

#### Chemical Extraction Methods Applied to Brown Seaweeds

Chemical extraction methods include pH shift protocols that employ a number of pH ranges to extract protein from seaweed biomass. Vilge et al. utilised a pH shift protein isolation protocol on *Saccharina latissimi*. The seaweed biomass was initially incubated at neutral pH followed by sonication and readjustment to alkaline pH (NaOH). The pH shift from neutral to alkaline pH greatly increased the precipitation of protein, whereby a result of 9% protein based on dry weight was obtained [64]. Additionally, Kumar et al. noted similar protein levels (8–12.2%) in *Sargassum wightii* using alkaline conditions [65]. Kadam et al. utilised acid (pH 2) and neutral (pH 7) conditions in the extraction of protein. This method reported protein yields of 7.97% (pH 2) and 16.9% (pH 7) from *Ascophyllum nodosum* on a dry weight basis [66]. Garcia-Vaquero et al. utilised ultra-sonication and ammonium sulphate precipitation, resulting in a protein yield of 6.5% protein from *Himanthalia elongata* [67].

#### Enzymatic Extraction Applied to Brown Seaweeds

Enzyme-assisted extraction has been reported to increase the total protein yield and the release of bioactive peptides from *Sargassum* sp. and *Laminaria japonica* [68,69]. A number of enzyme extraction methods have been attempted; for example, by Borines et al., who reported a protein yield of 10.2% from *Sargassum* spp. utilising an enzyme-assisted extraction approach with enzymes cellulose and β-glucosidase [68]. Additionally, Je et al. reported protein yields of between 6.94% and 22.52% from *Laminaria japonica* using a variety of carbohydrases including AMG, Celluclast, Dextrozyme, Promozyme^®^, and viscozyme^®^ and proteases such as alcalase^®^, flavourzyme^®^, Neutrase, Protamex^®^, and Pepsin enzymes [69].

#### Bioactivities of Peptides from Brown Seaweed Hydrolysates

The bioactivities derived of peptides from brown seaweeds include anti-hypertensive, anti-bacterial, and antioxidant activities shown in Table 3. Brown seaweeds have been known to inhibit ACE-1 and produce a hypotensive effect. Sato et al. reported a number of peptides that inhibited ACE-1 that were derived for a protease S “amano” hydrolysate of *Undaria pinnatifida*. Peptide sequences identified included VY, IY, AW, FY, VW, IW and LW. The VY and VW peptides were particularly effective in lowering systolic blood pressure in spontaneous hypertensive rats by −19 and −22 mmHg, respectively [114]. Suetsuna et al. also found ACE-1 inhibitory peptides from *Hizikia fusiformis*. The peptides were SKTY, GKY and SVY. These produced an anti-hypertensive effect in spontaneous hypertensive rats [115]. Other brown seaweed hydrolysates from which ACE-1 inhibitory peptides were derived previously include *Laminaria japonica* hydrolysates generated with alcalase^®^ (EC 3.4.21.14), papain (EC 3.4. 22.2), and trypsin. Chen et al. identified nine peptides (KY, GKY, STKY, AKY, AKYSY, KKFY, FY, and KFKY) that inhibited ACE-1 from this species, with the highest inhibitory activity observed for the peptide AKYSY (IC_50_ −20.63 μM) [117]. Zheng et al. reported an anti-hypertensive peptide from papain and pepsin hydrolysate of *Sargassium maclurei* (RWDISQPY peptide sequence) [119]. Anti-bacterial activity has been reported for peptides identified from *Saccharina longicruris*. Beaulieu et al. isolated several peptides that restricted the growth of the bacterium *Staphylococcus aureus* [116].

#### Application of Brown Seaweeds in Foods and Feeds

Brown seaweeds are used in the food and feed industries [146]. For example, *Laminaria* sp. provides a functional ingredient in fish feeds like AquaArom^®^ that has antioxidant effects and reportedly promotes growth of Atlantic salmon [126]. Brown seaweed has been used in a number of food products including meat and baked products, e.g., *Himanthalia elongata* has been used in the food industry as a functional ingredient in meat products such as pork patties and sausages to enhance the flavour, texture, taste, appearance as well as extending shelf life [122,148,149,150,151]. Moreover, *Himanthalia elongata* was used previously to enhance the phytochemical content of wholemeal and white wheat flour [122]. The higher phytochemical content, including that of flavonoids, is advantageous as these compounds have antioxidant effects when consumed through products including seaweed breadsticks [122]. Additionally, Arufe et al. demonstrated an improvement in the antioxidant properties of white bread following the addition of powdered *Fucus vesiculosus*, whereby density and crumb texture were improved significantly, therefore demonstrating the applicability of seaweed in human food products [123]. Seaweeds have been used to enhance the nutritional profile of carbohydrate foods such as pasta and dairy products [124,125]. O’Sullivan et al. indicated the improvement in the nutritional profile of milk and the shelf life of the milk when *Undaria pinnatifida* and *Fucus vesiculosus* seaweeds were incorporated as functional ingredients in milk [125]. Additionally, Prabhasankar et al. utilised *Undaria pinnatifida* as a functional ingredient that increased the carbohydrate and fat content of pasta through the enhancement of the gluten network when seaweed was incorporated [124].

### 1.5. Nutritional Composition of Seaweeds

Seaweeds are comprised of different vitamins (A, B, C, D, E, and K), minerals (zinc, iron, and magnesium) and amino acids including aspartic acid, glutamic acids, alanine, histidine, arginine, leucine, threonine, and minerals, which are essential for human health [11,68,73,152,153,154]. Different species, locations, and seasons of harvest result in different nutritional composition of protein. The protein content of seaweeds varies from 3 to 47% by dry weight depending on the type of seaweed [11,30]. The red seaweeds are the most concentrated in protein content, with the mean amount of protein between 10 and 30% of its dry mass and with some species including *Porphyra* and *Pyropia* spp. being reported as having higher values (47%) [139,155,156]. The protein content of green seaweeds ranges from 10% to 30% [37], with Khairy et al. reporting a protein content of 16.7–20.1% dry mass from *Ulva latuca* [156]. The protein content of brown seaweed is low at 3–15% dry weight [93].

Some studies have reported that seasonal differences in protein content in different types of seaweed can be attributed to water motion as simulating wave action can be a key regulatory factor in the uptake of nutrients and the growth of seaweeds [157,158]. However, this can depend on seaweed morphology and growth form as dense beds act as an additional layer in which nutrients are expected to cross via active transport/facilitated transport and or passive diffusion [159]. Reductions in protein content have been seen in brown seaweeds, as Schiener et al. reported a higher protein content of *sargassum latissima* at 9.9% dry weight during spring and lower protein contents of percentage dry weight during summer. 

This pattern was repeated in the case of other brown seaweeds including *Laminaria digitata*, *Laminaria hyperborean*, and *Alara esculenta* [160]. Additionally, Mohy El-Din et al. reported seasonal variations in different seaweeds, as *Ulva lactuca* and *Corallina mediterranea* collected in spring months reported the highest protein content at 23.2% and 20.6%, respectively [147]. Their proposed explanation was that greater nitrogen and phosphorus uptake occurs during winter and spring months, whereby greater uptake of these nutrients was used to sustain growth of seaweeds during the summer months [147,161,162].

Seaweeds are composed of a number of polysaccharide components including ulvan, alginic acid, laminarin, and fucoidan, which are found in specific types of seaweed including green (ulvan) [163], brown (alginic acid, laminarin, and fucoidan), [164] and red (agar and carrageenan) [165]. Moreover, the composition of polysaccharides within seaweeds can be different due to factors such as location, time of harvest and species [166,167]. Sfriso et al. identified seasonal changes in the composition of different sulphated polysaccharides in *Gracilaria* sp. and *Ulva rigida*. It was shown that *Gracilaria vermiculophylla*, *Gracilaria longissimi*, and *Gracilaria gracilis* exhibited maximum production of agar between March and August while *Ulva rigida* reached seasonal maximum production of ulvan in June [168]. Additionally, Skriptsova et al. reported different compositions of fucoidan from several brown seaweeds including *Saccharina japonica* and *Sargassum pallidum*. The maximum production of fucoidan was reported in *Saccharina pallidum* during February to July and *Saccharina japonica* in September to October [169]. Therefore, it can be comparatively shown that the accumulation of polysaccharide components within seaweeds including ulvan, agar, and fucoidan occurred mainly during the summer months with the exception of *Saccharina japonica* compared to the protein content that has reported the highest peaks during spring [147].

Seaweeds are a reliable source of macronutrients including important minerals (e.g., calcium, zinc, phosphorus and iron) that are essential to human nutrition. Seaweeds have exhibited high capacities for bioadsorptive and bioaccumulative abilities [170]. Owing to these properties, several minerals including iron, and potassium have been shown to be present at 10 to 100-fold higher in seaweeds than in traditional crops [170]. Higher iron content has been observed brown seaweed species; for example, Neto et al. have reported an iron content of 211 mg/100 g dry weight for *Gracilaria* spp. collected from Portugal, followed by 185.4 mg/100 g dry weight for *S. latissima* collected from northern France [171]. Additionally, Afonso et al. reported iron content in red seaweed species *Agarophyton vermiculophyllum* at 96.4 mg/100 g dry weight [172]. These results are significant compared to terrestrial crops such as sweetcorn, which has been reported to have an iron content of 4.9 mg/100 g dry weight [170]. Therefore, seaweed could potentially act as a supplement of iron, with the recommended daily consumption being 10–18 mg [173].

Calcium, zinc and phosphorus can be found in abundant quantities in seaweeds [174]. Typically, red species have a reported lower calcium content compared to that of brown and green seaweeds [175]. In particular, Parjikolaei et al. reported exceptionally high amounts of calcium from green seaweed species *Chondrus crispus* at 5398 mg/100 g dry weight [176]. Likewise high calcium content has been reported by Ometto et al., who noted a seasonal high of calcium in *Alaria esculenta* at 3100 mg/100 g wet biomass [177]. Variable amounts of zinc have been identified in a number of macroalgae; however, zinc is most prevalent in brown and red algae. Wallenstein et al. successfully harvested *Fucus spiralis*, *Porphyra* spp., and *Gracilaria* spp. from Portugal and reported a zinc content of 15 to 740 mg/kg dry weight, 10 to 82 mg/kg dry weight, and 24 to 163 mg/kg dry weight, respectively [178]. Red seaweeds are particularly rich in phosphate content, with examples including *Ulva* spp., *C. crispus*, and *Gracilaria* spp. Parjikolaei et al. reported a potassium content of 289 mg/100 g dry weight, and 109 mg/100 g dry weight for *Chondrus crispus* and *Gracilaria* spp., respectively [176]. 

Polyphenolic compounds are considered anti-nutritional due to their ability to inhibit animal and plant nutrition as they are known to bind to protein which reduces their bioavailability of the nutrient upon consumption [91,179]. Tyrpsin inhibitors are found in anti-nutrition factors, such as legumes reducing the absorption of proteins through the formation of two different protein complexes, directly resulting from the inhibitory action of pancreatic enzymes trypsin and chymotrypsin [180]. Phenolic acids are present in all types of seaweed and are active in essential processes including photosynthesis, protein synthesis and nutrient absorption as an antioxidant. There are various types of phenolic compounds including flavonoids, terpenoids, and tannins [153,181]. Red and green seaweed contain the largest number of phenolic compounds (flavonoids and bromophenols), followed by brown (phlorotannins) [181]. Phenolic acids have attracted notable interest due to their bioactivities components, which that have been reported to include anti-microbial, anti-viral, anti-diabetic, and antioxidant activities [182]. The antimicrobial properties of phenolic acids may contribute to the reduced bioavailability of protein, as the mode of action for phenolic acids (e.g., phlorotannins) is the formation of a complex containing available cellular metal ions [183]. Lack of freely available cations such as Na^+^, K^+^, Mg^2+^, and Ca^2+^ prevents the folding of messenger RNA molecules, leading to reduced protein formation in the ribosome, thereby affecting protein synthesis within seaweeds [183,184,185].

### 1.6. Digestion and Bioavailability of Proteins

Protein digestion involves the breakdown of protein from its complex molecular structures into smaller peptide fragments in vivo, requiring the utilisation of numerous enzymes and hormones within specific pathways of digestion [186]. The digestion of protein is initiated following the ingestion of food that enters the gastrointestinal tract via the oesophagus to the second stage of digestion within the stomach [187]. The proteolytic enzyme pepsinogen is secreted with hydrochloric acid (HCl) in the stomach and is converted into its active form, pepsin, which catalyses the hydrolysis of proteins into smaller polypeptide fragments. As the food moves into the intestine, enterokinase enzyme is secreted, resulting in the formation of the pro hormone trypsin, which binds to protein at residues lysine or arginine that in turn cleave the C-terminal generated free amino acids and peptide fragments [188]. The generated peptide fragments interact with membrane-bound peptidases and are digested into dipeptides or tripeptides. The peptides enter the enterocyte luminal membrane, resulting in the formation of individual amino acids through the activity of cytosolic peptidases, passing from the basolateral membrane into the bloodstream, making protein available for functional uses within the body [189].

Phenolic compounds including phytic acid, lectin, and tannins reduce the availability of nutrients such as protein [91,190,191]. Phenolic compounds are secondary metabolites in seaweed that act as components in seaweed cell walls [192], are soluble in water and are bound to proteins and polysaccharides [193]. In addition to phenolic compounds, a number of polysaccharides make up the cell walls of seaweed including alginic acid, laminarin, fucoidan, and carrageenan [31,164,194]. These polysaccharides cannot be digested by humans who do not have the enzymatic capabilities to digest and degrade the (1-4)-β-D-glycosidic bonds. Additionally, seaweed proteins are bound intercellularly via ionic bonds within the cell wall, making them inaccessible to gastrointestinal enzymes, and therefore the complexities of the seaweed cell walls have proven to be a major drawback towards a process for protein extraction [195,196]. However, there have been advances in the extraction process that have demonstrated potential in overcoming the cell wall complex as some trialled methods utilising unique profile of proteolytic and carbohydrase enzymes to disrupt the bonds between the cell wall and phenolic compounds, resulting in increased protein yields and the isolation of bioactive peptides exhibiting bioactivities such as anti-diabetic, anti-hypertensive, anti-inflammatory and antioxidant activities [12,54,55,56,103,107]. Although the use of seaweed-derived proteins in food applications, e.g., functional food and animal feeds, is limited, improvements in the purification process and additional research in the bioactivities of these peptides may contribute to the potential recognition of these components in a human food application [15].

A number of in vitro studies have highlighted the role of dietary fibre, as well as anti-nutritional factors in the digestion of seaweed-derived proteins. Urbano et al. noted a decrease in the digestibility of protein derived from *Porphyra tenera* and *Undaria pinnatifida* using a Wistar rat animal model that indicated that dietary fibre may alter the digestibility of seaweed proteins [197]. Goñi et al. analysed several indigestible seaweed-derived proteins from brown seaweeds. *Fucus vesiculosus*, *Laminaria digitata* and *Undaria pinnatifida* and red seaweeds *Chondrus crispus* and *Porphyra tenera* were tested for digestibility in Wistar rat caecal droppings. It was noted that all seaweeds contained significant amounts of unavailable protein, ranging from 2 to 24% [198]. Moreover, Fleurence et al. reported that the presence of phenolic compounds in a green seaweed, *Ulva armoricana*, might play a contributing role in in vitro protein digestibility as well as high polysaccharide content [199].

There are a number of methods that can be used to determine in vitro protein digestibility including animal and cell cultures, solubility, and dialysability models [200]. Animal and cell in vitro cell cultures can utilise a variety of cells including Madin–Darby canine kidney cells (dog kidney cells) as well as human colon adenocarcinoma (Caco-2 cell, HT29 cells) [201,202]. However, Caco-2 cell line models are typically used in digestive studies because the initial monolayer contains digestive enzymes active in the GI tract including peptidases, disaccharides and the resulting differentiated layer of enterocyte cells resulting from Caco-2 cells lines which are structurally similar to those of the intestinal epithelial layer [203]. Due to the number of in vitro digestion methods available that operate using different parameters including the different stages of digestion [204], the composition of electrolytes and enzymes and the mechanical forces applied indicated inconsistent digestibility results [205,206]. Therefore, the European Cooperation in Science and Technology (COST) has helped develop standardised digestibility models, through the utilisation of food matrices, nutraceutical delivery systems and simulating physiological conditions in the GI tract [204].

These digestibility models aim to simulate the conditions, e.g., pH, temperature, specific enzyme activity and agitation, experienced during digestion in vivo. There are two specific types of dialysability-based methods—static and dynamic. Static systems are the simplest, whereby products are immobilised within a single container. Processes including oral, gastric, and intestinal digestion are simulated via homogenisation, centrifugation and/or filtration. These models are primarily used in studies related to simple foods and isolated or purified nutrients [207]. Dynamic models are highly regulated computer-based bioaccessibility systems that tightly monitor parameters such as temperature, pH, as well as mixing and residence times within different compartments in order to provide a more accurate model of digestion and bioaccessibility [208].

### 1.7. Health Benefits of Seaweed Bioactive Peptides

#### 1.7.1. Angiotensin-I-Converting Enzyme (ACE-1; EC 3.4.15.1) Inhibition

Hypertension is a risk factor for cardiac-related diseases, presently affecting approximately 972 million people worldwide, and is predicted to include 1.56 billion by the year 2025. Currently, there are only a few treatment options available including drugs and the induction of heart stents combined with a healthy diet consisting of reduced lipoprotein (LDL), resulting in reduced blood pressure [209]. ACE is one of the key regulatory enzymes in the renin–angiotensin–aldosterone system (RAAS), which is responsible for the increase in blood pressure and heart function along with another key enzyme renin [209]. Renin is produced in response to low blood flow and within the RAAS system, renin acts on angiotensinogen, resulting in the conversion angiotensin I into angiotensin II by the action of angiotensin-I-converting enzyme [112]. Angiotensin II has numerous biological effects in the kidneys, adrenal cortex, arterioles, and brain, whereby it binds to AT1 receptors as shown in Figure 3 [210].

In the renal system, angiotensin II simulates the transport of sodium in the proximal tubules via AT1 receptors in the opening of sodium channels and urea transport in the inner medulla, which enhances the reabsorption of sodium and water within the body [211]. Angiotensin II induces the production of aldosterone in the adrenal cortex and the brain, resulting in increased thirst and generation of extracellular fluid volume with increased blood pressure, a common indicator of cardiovascular disease [212,213,214,215]. Moreover, high levels of angiotensin I can lead to increased vasoconstriction through the action of the second messenger 1,4,5-inositol trisphosphate (IP_3_), resulting in the formation of extracellular calcium, leading to elevated blood pressure [216,217]. However, this can be prevented by the action of ACE that cleaves and inactivates a peptide called bradykinin that decreases blood pressure [218].

There are a number of prescribed synthetic drugs such as Captopril^®^, Vasotec^®^ under the tradename Enalapril, Alcacepril, and Lisinopril that are utilised to inhibit the activity angiotensin-I-converting enzyme [209]. These drugs have side effects including hypotension, dry cough, and impaired renal function [54]; therefore, it would be advantageous to find a natural alternative which would avoid unwanted side effects. An example of this includes ACE-1 inhibitors such as bioactive peptides from species of red, green, and brown seaweeds [112,114,219]. The human consumption of seaweeds has been a part of Asian cultures for decades [220] both directly and in the form of functional food ingredients [221]. It has been linked to a decrease in the incidence of coronary heart disease in Japan [222].

There are a number of different types of seaweed such as *Undaria pinnatifida* (wakame), *Palmaria palmata* (dulse), *Porphyra yezoensis,* and *Sargassium maclurei* that have shown inhibitory activity for ACE (Table 2). Sato et al. identified a number of dipeptides resulting from protease hydrolysates from *Undaria pinnatifida* (wakame). Four of these peptides (VY, IY, FY, and IW) were administered at 1 mg/kg in spontaneously hypertensive rats (SHR), and indicated strong in vitro ACE-1 inhibitory activity, resulting in a reduction in blood pressure compared to the standard control Captopril [114]. Furuta et al. documented a number of ACE-1 inhibitory peptides, VYRT, LDY, LRY, and FEQDWAS, from *Palmaria palmata*. The peptides were generated using thermolysin enzymes in conjunction with pepsin-trypsin-chymotrypsin digests, with the highest ACE-1 inhibitory activity with an IC_50_ value of 6.1 μM for the LDY peptide [107].

*Porphyra yezoensis*-derived peptides have exhibited inhibitory activity for ACE, as Qu et al. carried out several studies which demonstrated anti-hypertensive activity from alcalase^®^ (EC 3.4.21.14) hydrolysates with an IC_50_ (the concentration of inhibitory peptides required to inhibit 50% of the ACE-1 enzyme) value of 1.6 and 0.516 g/l, respectively [108,109]. Brown seaweed *Sargassium maclurei* has been documented to inhibit ACE-1, as Zheng et al. identified a peptide (RWDISQPY) from hydrolysis with papain and pepsin (EC 3.4.23.1) enzymes with an IC_50_ value of 72.24 μM [119]. Therefore, seaweed has the potential to inhibit ACE-1 similar to other known synthetic drugs including Captopril^®^ and Enalapril and could potentially serve as an alternative treatment for high blood pressure.

#### 1.7.2. Renin (EC 3.4.23.15) Inhibition and Its Role in the Prevention of Hypertension

As previously mentioned, renin is one of the two key enzymes along with ACE within the renin–angiotensin–aldosterone system (RAAS) that is responsible for the increase in blood pressure, considered a risk factor in the generation of hypertension [209,225]. Renin is a highly specific aspartyl protease enzyme produced within the juxtaglomerular cells of the kidneys with only one known substrate angiotensinogen (AGT). The enzyme’s inactive precursor form, prorenin, is stored in the vesicle cells and is converted into renin in response to specific stimuli such as extracellular fluid volume and blood pressure within the RAAS system [226,227]. Following these events, angiotensin I is converted into angiotensin II by the action of angiotensin-I-converting enzyme [114], which results in the generation of increased blood pressure and hypertension [228]. Angiotensin-I-converting enzyme (ACE-1) cleaves and inactivates a peptide called bradykinin, a stimulator of vasodilator prostaglandins, resulting in reduced blood pressure [229,230].

Renin inhibition is an active clinical strategy in the control of blood pressure and hypertension that has been achieved through the use of renin inhibitors such as Aliskiren, otherwise referred to as Tekturna^®^ or Rasilez^®^ [231], whose action inhibits the activity ACE-1, thus preventing the conversion of angiotensin I to angiotensin II [232]. Unwanted side-effects can occur as a result of these agents including erectile disfunction [233,234], dry cough [235] and congenital malformations [236,237]. A potential alternative to pharmacological intervention is to utilise bioactive peptides derived from seaweeds which have displayed inhibition of both renin and ACE-1 [238]. A number of bioactive peptides derived from red and green seaweeds have displayed inhibition of renin [6,110]. Fitzgerald et al. identified a chemically synthesised propeptide (IRLIIVLMPILMA peptide sequence) resulting from papain (EC 3.4. 22.2) hydrolysis of *Palmaria palmata* that indicated 58.97% renin inhibition when compared to the positive control (Z-Arg-Arg-Pro-Phe-His-Sta-IleHis-Lys-(Boc)-OM) [110]. A green seaweed species, *Ulva lactuca*, has also indicated renin inhibitory activity. Garcia-Vaquero et al. colleagues reported that three ultra-filtered papain hydrolysate fractions (1 kDa-UFH, 3 kDa-UFH and 10 kDa-UFH) inhibited renin in varying degrees (6.89–21.19%) [6], which is significantly lower compared to the *Palmaria palmata* propeptide in the Fitzgerald et al. method [110]. Therefore, the screening of further seaweeds for renin inhibition would be beneficial as red and green seaweeds could be utilised in the generation of natural renin inhibitors.

#### 1.7.3. The Role of Platelet-Activating Factor Acetylhydrolase (PAF-AH)

Platelet-activating factor acetylhydrolase (PAF-AH; EC 3.1.1.47), also known as lipoprotein-associated phospholipase A2, is mainly produced by macrophage cells following maturation from initial monocyte cells [239,240,241]. Its active form is secreted in the plasma of the blood, whereby it binds to lipoproteins such as low-density lipoproteins and high-density lipoproteins in humans [242,243]. In addition, there are a number of tissues in the body that PAF-AH can be found in including the aorta mammary and prostate glands as well as the liver and intestine [244,245,246,247]. PAF-AH secretion is regulated through a number of cytokines, anti-inflammatory agents and steroid hormones [248]. Platelet-activating factor acts a mediator for a number of mediator inflammatory processes including angiogenesis, thrombosis, carcinogenesis, metastasis, arthritis, vascular disease, and edema [249,250,251,252]. Particular attention has developed around PAF-AH and its role in the cardiovascular disease atherosclerosis, which is an inflammatory disease of the arterial wall, resulting in strokes [253,254]. PAF acetylhydrolases are bound to apolipoprotein β lipoproteins and act as mediators in atherosclerosis, which promotes lipid oxidation [255,256].

During lipid oxidation, lipids release biologically active by-products such as lysophosphatidylcholine and oxidised non-esterified fatty acids (oxNEFAs), and the recruitment of monocyte/macrophages into the atherosclerotic lesion causes the blockage and the deterioration of heart conditions such as stenosis and the thrombotic occlusion of arteries connected to the heart, brain, and legs as shown in Figure 4 [257]. Therefore, inhibition of PAH-AH would be beneficial in preventing the onset of diseases such as cancer and atherosclerosis [258].

Currently, synthetic statin drugs are prescribed to limit cholesterol, preventing the build-up of lower-density lipoprotein; however, they have documented side effects such as statin-associated muscle symptoms, neurological and neurocognitive effects, hepatotoxicity, and renal toxicity [259]. There have been issues in the development of PAF-AH inhibitors in the treatment of atherosclerosis, as GlaxoSmithKline’s drug Darapladib failed to pass beyond its second phase III (STABILITY) clinical trial due to a reported lack of prevention in major coronary events and the generation of unexpected side effects including diarrhoea, malodorous faeces, urine, and skin [260]. Additionally, increased PAF-AH activity has been associated with the generation of digital ulcers [261]. Due to the failure of Darapladib and other viable PAF-AH inhibitors, statin drugs still remain the ideal choice of clinical treatment as they have been reported to inhibit plasma PAF-AH levels as well as cause a reduction in cardiovascular events [262]. The development of an alternative natural therapy that would inhibit PAF-AH would be ideal and an example of this is seaweed.

Moreover, Fitzgerald et al. identified a tetra-peptide (NIGK peptide sequence) derived from papain (EC 3.4. 22.2) hydrolysates of *Palmaria palmata* that indicated a PAF-AH with an (IC_50_) at 2.32 mM compared to the positive control methyl arachidonyl fluorophosphonate (MAFP) [111]. Therefore, this may represent a potential alternative choice in therapy if more seaweed species were screened, although this would have to be based on potential toxicological and allergic effects associated with the specific seaweed.

#### 1.7.4. α-Amylase and α-Glucosidase Activity in Diabetes Mellitus

Pancreatic enzymes, such as α-amylase (EC 3.2.1.1), represent one of the key enzyme groups present in the small intestine which play an important role in the hydrolysis of starch, e.g., to maltose, maltotriose, and other simple monosaccharides in the digestive system [263]. α-glucosidase (EC 232-604-7), an enzyme present in the small intestine, catalyses the glycosylic bonds of these oligosaccharides, resulting in the absorption of glucose in the bloodstream [264,265,266]. In certain cases, this increase in blood sugar levels may contribute to the loss of regulation of glucose in the blood, resulting in the onset of type 2 diabetes mellitus [267,268]. Therefore, an attractive therapeutic strategy in the treatment of type 2 diabetes is the inhibition of the activity of α-amylase and α-glucosidase [269] as increased α-amylase activity has been noted as a contributing factor in the generation of high post-prandial levels of glucose [270]. There have been a number of established therapeutic agents that have been developed in order to inhibit the action of α-amylase and α-glucosidase, with Acarbose, Miglitol, and Voglibose being the most commonly prescribed drugs in the inhibition of carbohydrate digestion and absorption [271,272]. Seaweed-derived peptides have been reported to exhibit α-amylase and α-glucosidase inhibitory activity [112,118]. A number of seaweed-derived peptides have reported inhibitory activity, as Admassu et al. reported in the inhibition of α-amylase with two peptides (GGSK, ELS) derived from hydrolysates of *Porphyra* sp. [112]. Additionally, Hu et al. identified two peptides derived from *Spirulina platensis* that displayed α-amylase and α-glucosidase inhibition [118].

Therefore, seaweed has indicated the ability to inhibit α-amylase and α-glucosidase inhibitors that could potentially serve as a potential alternative to bacterially derived inhibitors.

#### 1.7.5. Cyclooxygenase (COX) Enzyme Inhibition and Role in Prevention of Inflammation

Inflammation is the process directed by the immune system in response to situations such as infection, injury, and diseases [273]. This prevents further damage to the host tissue through the trapping and destruction of target molecules, while directing the generation of new tissues and restoring physiological functions within the body [274]. A key molecule in the generation of immune responses is prostaglandins which are molecules that sustain and mediate key inflammatory responses including the regulation of pain [275]. Cyclooxygenase (COX) enzymes (EC 1.14.99.1) play a pivotal role in the inflammatory response in humans through the conversion arachidonic acid into lipid autacoids prostaglandins and thromboxane in the arachidonic acid metabolic pathway as shown in Figure 5 [276]. Arachidonic acid is an omega-6 polyunsaturated fatty acid acquired by humans and mammals through the consumption of fish, eggs and lean meat [277,278] and binds to the phospholipids in the cell membrane of the brain, spleen, liver and retina of the human body [279,280]. COX enzymes are monotopic membrane-bound proteins that can be produced in two forms in humans, COX-1 and COX-2, with arachidonic acid acting as the substrate [281,282].

The two isoforms of cyclooxygenase have specific functions in the body; COX-1 regulates the control of renal blood flow and initiates protection from stomach ulcers and the production of thromboxane in platelets and prostaglandin E2 (PGE2) molecules, resulting in the maintenance of ideal homeostasis conditions [283]. COX-2 induces the production of prostaglandin molecules which are active in the inflammation process as well as in proliferation diseases such as cancer [284]. A number of molecules can stimulate the production of COX-2 including cytokines, hormones including lipopolysaccharide (LPS), interleukin-1 (IL-1), platelet-activating factor (PAF) as well as tumour promoters such as tumour necrosis factor (TNF) [285,286,287]. Therefore, it has been indicated that the overexpression of COX-2 can result in the resistance of apoptosis, leading to the proliferation, angiogenesis, and metastasis of tumour cells [288,289,290].

The inhibition of the cyclooxygenase enzyme can be achieved using non-steroidal anti-inflammatory drugs (NSAIDs), resulting in anti-inflammatory effects in the treatment of diseases such as ischemic heart disease and cancers including colon cancer [291,292]. There are several established COX inhibitors in the form of non-steroidal anti-inflammatory drugs (NSAIDs) such as celecoxib, etoricoxib, and lumiracoxib that have been used in the management of conditions such as arthritis [293,294]. There is a growing possibility of new therapeutic targets including cancer, as well as Alzheimer’s disease and Parkinson’s disease [295]. Seaweed-derived peptides have reported COX inhibition including *Pyropia yezoensis* [113]. Lee et al. has indicated COX inhibitory activity with a bioactive peptide (PPY1) derived from *Pyropia yezoensis* using a mouse lipopolysaccharide (LPS)-stimulated macrophage cell line RAW 264.7. When the PPY1 peptide was applied to the macrophage cell line, it resulted in the suppressed release of cytokine, cyclooxygenase-2 (COX-2). No significant cytotoxicity was reported when tested in a cell viability assay over 24 h [113]. This indicates that seaweed-derived peptides have the potential be used in the treatment of pathogenic diseases, e.g., cancer and inflammatory diseases, by supressing the COX-2.

### 1.8. Effects of Food Processing on the Availability of Proteins and Bioactive Peptides

Food processing uses processes such as dehydration, thermal heating, or fermentation that can cause damage to functional foods and animal feeds that contain seaweed-derived protein and bioactive peptides [300,301,302,303]. Some of these processes, e.g., heating, can be beneficial in terms of protein digestibility, which enhances emulsification properties [304]. However, too much heating can also impact negatively on essential amino acid bioavailability; for example, non-enzymatic browning resulting from these processes can cause the condensation of amino acid groups such as lysine that is irreversibly modified into the N-substituted lysine in the protein structure, reducing the protein bioavailability when consumed [305,306,307].

Fermentation with lactic acid bacteria has been used to preserve the stable structure of proteins as well as generating bioactive peptides from the parent protein (LAB) [308,309]. Lactic acid can be produced through fermentation with microbes such as fungus *Rhizopus* [310,311], yeast *Pichia stipites* [312] as well as lactic acid bacteria that exhibit high proteolytic activity, resulting in the hydrolysation of proteins and the generation of peptides and free amino acids [308,313]. The use of lactic acid bacteria as the substrate in the fermentation process has led to the identification of bioactive peptides, as Nakamura et al. positively identified angiotensin-I-converting enzyme (ACE) inhibitory peptides from fermented sour milk with the starter culture of *Lactobacillus helveticus* and *Saccharomyces cerevisiae* [314].

They identified two anti-hypertensive tripeptides (Ile-Pro-Pro and Val-Pro-Pro) derived from casein hydrolysis, which indicated IC_50_ values of 9 and 5 µM, respectively [314]. The use of seaweeds as a substrate in the fermentation process represents an attractive alternative option to other terrestrial plants (e.g., corn and rice) [315,316] due to their rapid growth rate and cultivation in the aquatic environment [317], and their reported bioactive peptide potential [10,318].

Lin et al. demonstrated the potential use of seaweeds in fermentation utilising *Gracilaria* spp., *Sargassium siliquosum*, and *Ulva lactuca* species. The highest amount of lactic acid production, at a concentration of 19.32 g/L, occurred with *Gracilaria* spp., under optimum conditions (30 °C over 72 h) and using starter cultures *Lactobacilus acidophilus* and *Lactobacillus plantarum* [319]. Though there have been few reports detailing the widespread use of macroalgae as a source of microbial fermentation, this method indicates the potential of seaweed as a substrate in this process; however, further screening for more seaweeds needs to be carried out. An additional process that can be used to generate proteins and bioactive peptides is ultra-high-pressure processing involving the use of pressure ranging from 100 to 1000 MPa that can break ionic and disulphide bonds within protein and thus generate peptides that are low in terms of molecular weight [68,309]. Garcia-Mora et al. reported antioxidant and ACE-1 inhibitory activity from peptide hydrolysates derived from lentil protein using pressures varying from 100 to 600 MPa [309].

### 1.9. Challenges Associated with Protein Extraction Technologies and Future Prospects

The extraction of seaweed proteins in economically viable yields has been particularly challenging due to the anionic cell wall composed of polysaccharide components and the large variability in protein content that is due, in part, to species used, geographical location and season of harvest [80,320]. A number of novel technologies have been developed in response to this problem as previously mentioned [54]. Ultrasound sonication and enzyme-assisted hydrolysis techniques are particularly important as they boast the potential to be scaled up [56]. There are distinct advantages of using ultrasound extraction—it is relatively inexpensive, has a short processing time and is a low energy consuming technique; however, some protein degradation can occur depending on the settings used [10]. Moreover, enzyme-assisted hydrolysis has a number of advantages—hydrolysis is reported to increase protein yield and the isolation of bioactive peptides with associated health benefits. Further, it is thought to reduce the potential of seaweed proteins to cause allergy. However, enzyme processes can be expensive [40,41,54,55,56]. In relation to the reported bioactivities associated with seaweed-derived peptides, it is important that these bioactivities are confirmed in models and animal trials. This may also pose ethical concerns around the use of animals in food product testing. However, until in vitro models more closely mimic animal models, this is a necessary step if producing seaweed proteins and hydrolysates for potential health products, where in order to make a claim, the product must be proven to comply with existing legislation in Europe or America governed by the European Food Safety Authority (EFSA) or the FDA, respectively.

## 2. Further Prospective 

### 2.1. Industrial-Scale Technologies for the Extraction of Protein from Seaweeds

Ultrasound and MAE are available as protein disruption methodologies at industry scale as are methods including high-pressure processing (HPP). Membrane filtration technologies including ultrafiltration and diafiltration methods may be applied to seaweeds such as *Grateloupia turuturu* to extract phycobiliproteins known as R-phycoerythrin at the pilot plant scale. In this process, industrial-type polyethersulfone 25–30 kDa membranes are used [321]. Results indicate that 100% of R-phycoerythrin can be recovered along with 32.9% of other proteins found in the seaweed and antinutritional compounds. Labour is the main expense of this process as well as maintaining the integrity of the membranes [321]. If this process could be adopted for extraction and isolation of seaweed proteins and bioactive peptides could be rapidly increased, this would greatly enhance their potential use in pharmaceutical, cosmetic and industrial applications. Challenges associated with macroalgal extraction technologies include costs incurred for labour and cleaning as well as environmental hazards associated with chemicals used for membrane cleaning processes. Other challenges include sensory and taste challenges associated with end macroalgal protein products, i.e., colour can limit the application of some macroalgal proteins and hydrolysates.

### 2.2. Discussion

Seaweeds can be considered an alternative and sustainable source of nutrition including protein, vitamins, and minerals. In terms of proteins, some species of seaweeds have reported a higher protein content compared to the concentrations found in meat and dairy [4,5]. There is wide variation in the protein content in seaweeds, with red species reporting the highest protein content. However, there are a number of factors that can affect the protein content in seaweeds including specific environmental factors such as temperature, as well as season and location of harvested seaweed. The digestibility of proteins from seaweeds and other plant sources is particularly challenging compared to animal-derived proteins (e.g., casein and whey) that boast a higher digestibility capacity [195]. This is due to the presence of polysaccharide components that make up the anionic cell wall that are intracellularly bound to proteins making the extraction of protein difficult [199]. In addition, insoluble phenolic compounds in some seaweed species can be bound to cell wall polysaccharides via hydrophilic and hydrophobic bonding which protein structures attached to [56]. Therefore, sophisticated and efficient disruption of the algal cell wall is essential to the extraction of seaweed-derived proteins. Additionally, the lack of harmonisation in the digestibility models studied such as the static and dynamic systems has made in vitro analysis quite difficult due to the varied and unreliable results reported [203]. There has only been a limited number of in vivo studies detailing the protein digestibility of seaweeds which have primarily utilised crude seaweed extracts instead of seaweed protein extracts, meaning further screening needs to be carried out in order to determine the effects of seaweed proteins in vivo in addition to in vitro [322].

A number of bioactive peptides have been identified within the protein sequences of different seaweeds that have exhibited unique effects such as anti-hypertensive, antioxidant, anti-inflammation, and anti-cancer effects. The applicability of these peptides in the medical diagnostics, nutraceuticals and cosmetic sectors is currently being investigated [10]. Further characterisation of bioactive peptides is needed to determine their use as a potential drug target as well as drug carriers in pharmaceutical applications, in the treatment of obesity, hypertension, diabetes, and other metabolic syndromes [112].

There is an increasing demand for improvement in extractible protein yield [21] using novel extraction techniques as well as the isolation and further screening of sensitive bioactive compounds [27]. Several methods have been reported in the literature to enhance the extractible yields of protein from seaweeds. However, some of these methods have been shown to potentially damage thermosensitive bioactive peptides, for example ultrasound-assisted extraction [10,66]. In addition, conventional chemical/liquid extraction involves the use of toxic organic solvents that have a significant environmental impact [90]. Techniques such as pulse electric field and enzyme-assisted extraction may provide a better means of extracting proteins [10,44] that are also greener due to the minimal use of organic solvents. In particular, pulse electric field is a non-thermal and energy efficient technique which requires no solvent [81,323]. To date, it has mainly been used to extract lipids [324]; however, scaling up has been problematic [46]. Enzyme-assisted extraction has been used to increase the extraction yield of proteins as well as the isolation of bioactive peptides [54]. Food-grade enzymes are routinely used to degrade polysaccharide cell walls and they require minimal use of organic solvents. They also offer the potential to be scaled up [56]. The potential improvement in the extraction, purification and isolation processes of seaweed protein will assist in the enhancement of potentially new bioactive peptides produced through novel techniques. These techniques are highly reproducible in terms of protein yield and are cost efficient when produced at large scale [10].

The sustainability of seaweed aquaculture is fundamental to the future use of seaweed proteins and their bioactive peptides, requiring the balance of potential environmental risk with economic reward [325]. This means using cultivation models that grow native species of seaweed which are harvested during the autumn and spring months [326]. The potential dangers of cultivating non-native species can have detrimental effects to the local ecosystem as the prevalence of pests and diseases in the ecosystem can pose a risk to large-scale seaweed cultivation due to the genetic diversity of wild seaweeds [327]. The sustainable cultivation of seaweeds as a source of dietary protein can potentially reduce the over-reliance of fishing through seaweed farming as well as adopting techniques such as micropropagation to facilitate the production of particular genotypes of seaweeds of high importance [328]. Currently, there are few products on the functional food market that utilise seaweed proteins, mostly being used for their antioxidant properties in human foods and animal feeds [8,11,139,140]. Therefore, increased research needs to be undertaken in order to promote the use of seaweed protein in the application of human foods such as functional foods.

In conclusion, seaweed has the potential to be used as a source of protein, vitamins, minerals, and most importantly as a source of bioactive peptides that can be used to promote health in humans. The establishment of efficient cell disruption techniques for algal cell walls will inevitably determine whether seaweeds can be used at a wider industrial scale in the food and pharmaceutical industry. To date, enzyme-assisted extraction has been successfully used to enhance protein yields and to generate bioactive peptides, which exhibit anti-hypertensive and anti-diabetic properties. However, further work is needed to improve yields and to screen other seaweed species for bioactivities that have physiological benefits in vivo.

## Figures and Tables

**Figure 1 foods-11-00571-f001:**
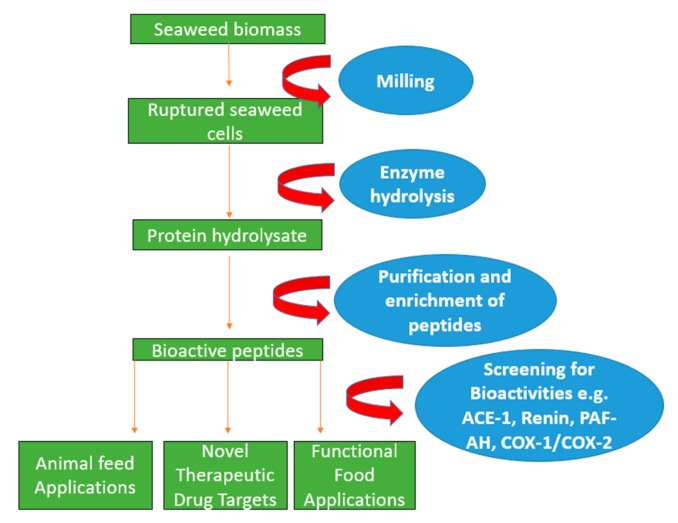
Schematic diagram indicating enzyme-assisted protein extraction of seaweed using hydrolysing enzymes to rupture the cell wall. This is then proceeded by purification and enrichment techniques such as filtration and molecular weight cut off in order to generate bioactive peptides that can be potentially used in applications in the pharmaceutical and nutraceutical sectors.

**Figure 2 foods-11-00571-f002:**
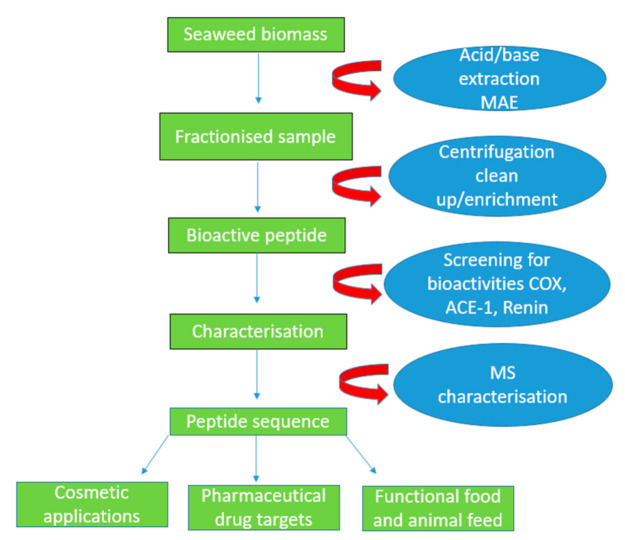
Schematic diagram indicating the pH shift and microwave-assisted extraction (MAE) procedures in order to disrupt the anionic cell wall in seaweeds. The acid/base is added, fractionising the sample into smaller fragments which are centrifuged and later cleaned up and filtered using a molecular weight cut-off filter membrane. The peptides are screened for anti-hypertensive, anti-diabetic and anti-inflammatory activities. Additionally, the peptides are characterised using mass spectrometry, and they can be used in the functional foods, as well as the cosmetic and pharmaceutical sectors.

**Figure 3 foods-11-00571-f003:**
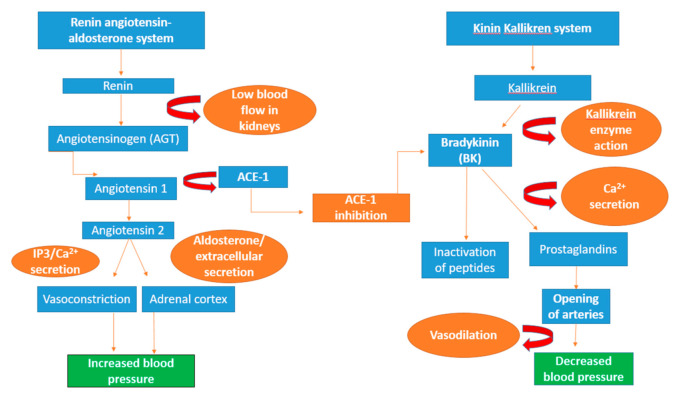
Schematic diagram of the renin–angiotensin–aldosterone system (RAAS), the kinin–kallikrein system and the anti-hypertensive effect of ACE. The protein angiotensinogen is formed in the liver and renin aspartic protease protein causes the breakdown of angiotensinogen into the decapeptide angiotensin I. Through the cleaving action of ACE-1, angiotensin I is converted into angiotensin II, whose action recruits the second messenger 1,4,5-inositol trisphosphate (IP^3^), resulting in the formation of calcium ions (Ca^2+^), leading to an increase in blood pressure through vasoconstriction. In the adrenal cortex, the secretion of angiotensin II enhances the production of aldosterone that causes the formation of extracellular fluid, thereby increasing blood pressure. Bradykinin a peptide produced through enzyme action of kallikrein, it is inactivated by ACE-1 and enhances the release of calcium ions [223]. Immune-regulating lipids and prostaglandins signal the opening of arteries in the heart through vasodilation, alleviating blood pressure [224].

**Figure 4 foods-11-00571-f004:**
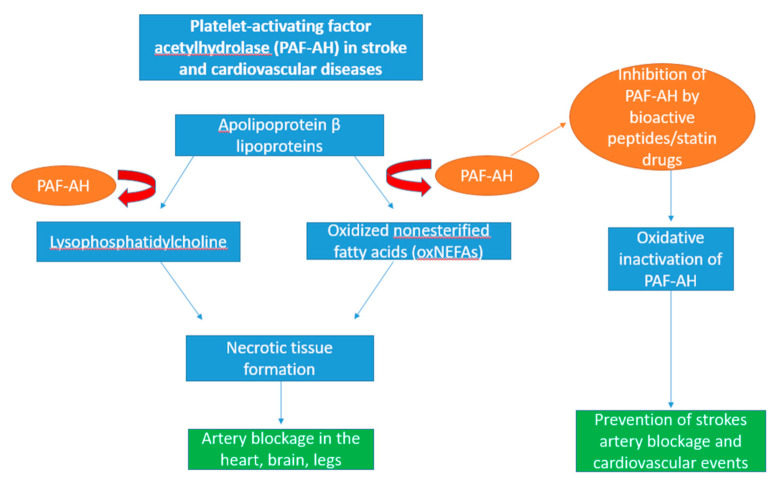
Schematic diagram showing the role of platelet-activating factor acetylhydrolase (PAF-AH) in cardiovascular disease. PAF-AH is a catabolic enzyme residing in immune cells such as macrophages and lipoproteins such as apolipoprotein β lipoproteins. Upon secretion of plasma PAF-AH, they actively oxidise apolipoprotein β-containing lipoproteins in the blood into two inflammatory mediators, lysophosphatidylcholine and oxidised non-esterified fatty acids (oxNEFAs). The formation of these molecules simulates the rapid production of diseased tissue which contribute to the blocking of arteries and transportation of blood to essential organs such as the heart and brain, leading to necrosis. Through the oxidation of PAF-AH, its activity can be inhibited by seaweed-derived bioactive peptides and statin drugs, thus preventing the onset of strokes and other cardiovascular events.

**Figure 5 foods-11-00571-f005:**
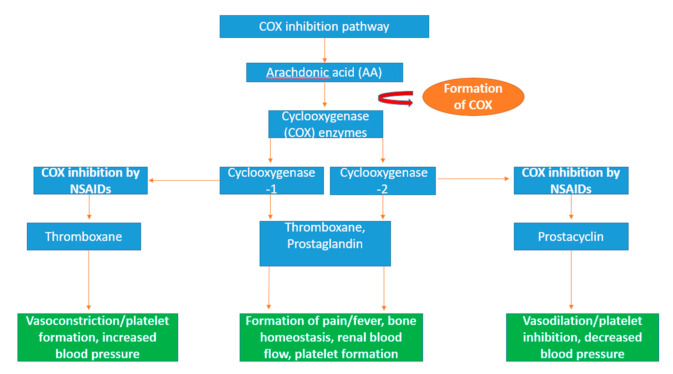
Schematic diagram of the arachidonic pathway and the inflammatory action of cyclooxygenase (COX) enzymes. Arachidonic acid is a polyunsaturated fatty acid which is converted into eicosanoid molecules prostaglandin and thromboxane by the rate-limiting enzymes, COX-1 and COX-2 [296]. The eicosanoid fatty acids prostaglandin and thromboxane are particularly involved in essential inflammation-related activities including the formation of bone and regulation of blood flow and platelets [297]. Non-steroidal anti-inflammatory drugs (NSAIDs) specifically target COX-1 and COX-2 activity. Eicosanoid vasodilator prostacyclin is targeted by NSAIDs, resulting in decreased blood pressure through vasodilation [298]. Thromboxane, another target of NSAIDs, increases blood pressure through vasoconstriction and increases the formation of platelets [299].

**Table 1 foods-11-00571-t001:** Protein extraction technologies applied to seaweeds to disrupt the cell wall.

Extraction Method	Principle	Advantage	Disadvantage	References
Enzyme hydrolysis	Proteolytic/carbohydrase enzymes are applied to degrade polysaccharide components within algal cell walls, releasing target proteins.	Limited use of organic solvents; specific and mild technique for extracting protein and bioactive components	Long extraction times; enzyme preparation can be expensive and require specific temperature and pH.	[29,39,40,41]
Microwave-assisted extraction	The extraction of target compounds occurs through the application of electromagnetic radiation, resulting in the breakdown of the bonds within the algal cell wall. Rapidly heating the sample solvent mixture results in wide-ranging applicability for the rapid extraction of analytes, including thermally unstable substances.	Environmentally friendly technique as it requires no organic solvents and short extraction times	Use in dried seaweed biomass may be limited.	[39,42,43]
Ultrasound-assisted extraction	The use of an acoustic cavitation technique to produce vapor bubbles in the extraction mixture that contributes to the disruption of polysaccharide components in seaweed biomass, thereby releasing proteins.	Fast processing time, low energy consumption, thermal sensitive technique, and limited use of organic solvents	Potential structural changes within polysaccharide structure.	[10,42,44]
Pulse electric field	The generation of high voltage facilitates protein extraction by electroporation, which disrupts the cell membrane of seaweed.	Non-thermal, energy efficient technique	Limitations on scaling up.	[44,45,46,47]
Solid–liquid extraction	Utilises different acid and alkaline conditions with water to facilitate the breakdown of hydrogen bonding in algal cell walls.	Simple and cost effective	Partial degradation of the proteins and bioactive components and time consuming.	[48,49,50]

**Table 2 foods-11-00571-t002:** Protein extraction methods applied to different seaweeds and reported protein yields (% dry weight).

Chlorophyta (Green)			
Seaweed Species	Extraction Methods	Protein Extracted (% Dry Weight)	Reference
*Ulva ohnoi*	Microwave extraction	23.9%	[58]
*Codium tomentosum*	Ultrasound extraction and enzyme extraction (with viscozyme^®^ L, cellulose (EC 232.734.4), alcalase^®^ (EC 3.4.21.14), favourzyme ^®^ (EC 232.752.2)	16.9−18.8%	[37]
*Ulva ohnoi*	Solvent extraction	12.28−21.57%	[59]
*Ulva* spp.	Enzyme extraction (with endopeptidase (EC 3.4.21.53), cellulase (EC 3.2.1.4), xylanase (EC 3.2.1.8), β-glucanase (EC 3.2.1.6), arabanase (EC 3.2.1.99)	24.4%	[34]
*Ulva Lactuca*	Enzyme extraction (with endopeptidase, cellulase, xylanase, β-glucanase, arabanase)	6.2–10.1%	[35]
*Ulva Lactuca*	Enzyme extraction (with papain (EC 3.4. 22.2))	69.19%	[6]
*Ulva Lactuca*	pH shift extraction (with pH 2 and pH 13)	22.7%	[49]
**Rhodophyta (Red)**			
**Seaweed Species**	**Extraction Method**	**Protein Extracted (% Dry Weight)**	**Reference**
*Palmaria palmata*	Autoclave	21.5%	[28]
*Porphyra umbilicalis*	Chemical extraction	22.5%	[49]
*Chondrus crispus*	Osmotic shock	35.5%	[28]
*Chondrus crispus*	Enzyme extraction (with cellulose)	7.1%	[60]
*Porphyra acanthophora*	Homogenisation and protein precipitation	8.9%	[61]
*Palmaria palmata*	Enzyme extraction (with Celluclast (EC 232-734-4), Shearzyme^®^, alcalase^®^, and viscozyme^®^)	35.5–41.6%	[12]
*Palmaria palmata*	Enzyme extraction (with Umamizyme^TM^ and xylanase)	33.4%	[62]
*Porphyra tenera*	Enzyme extraction (with Prolyve^®^1000, and flavourzyme^®^)	23%	[63]
*Chondracanthus chamissoi*	Enzyme extraction (with α-amylase, cellulose, and pectinase)	36.1%	[55]
**Phaeophyta (Brown)**			
**Brown Seaweed Species**	**Extraction Method**	**Extraction Yield (%DM)**	**Reference**
*Fucus vesiculosus*	High-pressure extraction	23.7%	[28]
*Fucus vesiculosus*	Autoclave extraction	24.3%	[28]
*Alaria esculenta*	High-pressure extraction	15%	[28]
*Alaria esculenta*	Autoclave extraction	17.1%	[28]
*Saccharina latissimi*	Chemical extraction	9%	[64]
*Sargassum wightii*	Chemical extraction	8–12.2%	[65]
*Ascophyllum nodosum*	Chemical extraction	7.97–16.90%	[66]
*Himanthalia elongata*	Chemical extraction	6.5%	[67]
*Sargassum* spp.	Enzyme extraction (with cellulose and β-glucosidase)	10.2%	[68]
*Laminaria japonica*	Enzyme extraction (with AMG, Celluclast, Dextrozyme, Promozyme^®^, viscozyme^®^, alcalase^®^, flavourzyme^®^, Neutrase, Protamex^®^, and pepsin (EC 3.4.23.1)	6.94–22.5%	[69]

**Table 3 foods-11-00571-t003:** Bioactivities reported previously for seaweed-derived bioactive peptides (IC 50 µM: the concentration of inhibitory peptides required to inhibit 50% of the enzyme).

Chlorophyta (Green)					
Seaweed Species	Extraction Process	Reported Bioactivity	Amino Acid Sequence	IC_50_	Reference
*Ulva lactuca*	Osmotic shock/precipitation with ammonium sulphate and papain (EC 3.4. 22.2) hydrolysis	Renin inhibition	None reported	None reported	[6]
*Ulva* sp.	Enzyme hydrolysis with purazyme, flavourzyme (E.C. 232-752-2)^®^, alkaline protease-Protex 6L,	Anti-inflammatory	None reported	None reported	[102]
*Ulva lathrata*	Enzyme hydrolysis with alcalase^®^ (E.C. 3.4.21.14), pH 7.6, 90 min, 25 °C//10 min, 100 °C	ACE-1 inhibition	PAFG	35.9 μM	[96]
*Ulva intestinalis*	Enzyme hydrolysis with trypsin (EC 3.4.21.4) + pepsin + papain, pH 8.42, 5 h, 28.5 °C	ACE-1 inhibition	FGMPLDR, MELVLR	219.35 μM, 236.85 μM	[95]
*Ulva rigida*	Enzyme hydrolysis with pepsin (E.C. 3.4.23.1) (E:S 1), pH 2, 20 h, 37 °C//bromelain (E:S 1), pH 7, 20 h, 37 °C	ACE-1 inhibition	IP, AFL	0.020 mg/mL, 0.023 mg/mL	[97]
*Ulva lactuca*	Osmotic shock/precipitation with ammonium sulphate and papain (EC 3.4. 22.2) hydrolysis	Renin inhibition	None reported	None reported	[6]
*Ulva* sp.	Enzyme hydrolysis with purazyme, flavourzyme (EC. 232-752-2)^®^, alkaline protease-Protex 6L,	Anti-inflammatory	None reported	None reported	[103]
**Rhodophyta (Red)**					
**Seaweed Species**	**Extraction Process**	**Reported Bioactivity**	**Amino Acid Sequence**	**IC_50_**	**Reference**
*Porphyra dioica*	Enzyme hydrolysis with alcalase^®^ (E.C. 3.4.21.14) and flavourzyme^®^ (E:S 1), pH 7, 4 h, 50 °C	DPP IV inhibitory activity	WLVA	439 μM	[10]
*Porphyra dioica*	Enzyme hydrolysis with alcalase^®^ and favourzyme^®^ (E:S 1), pH 7, 4 h, 50 °C	ACE-1 inhibition	DYYLR, AGFY, YLVA, TYIA	551 μM, 382.4 μM, 163.6 μM, 89.7 μM,	[10]
*Porphyra yezoensis*	Ion exchange and gel filtration method	ACE-1 inhibition	IY, MKY, AKYSY, LRY	2.69 μM 7.26 μM 1.52 μM 5.06 μM	[104]
*Porphyra yezoensis*	Enzyme hydrolysis with pepsin (EC 3.4.23.1), 5 h, 45°C	anti-coagulant activity	NMEKGSSSVVSSRM	0.3 µM	[105]
*Palmaria palmata*	Enzyme hydrolysis with corolase PP, pH 7,4 h, 50 °C	Antioxidant	FITDGNK, NAATIIK, ANAATIIK, SDITRPGGQM, DNIQGITKPA, LITGA, LITGAA, LITGAAQA, LGLSGK, LTLAPK, LTIAPK, ITLAPK, ITIAPK, VVPT, QARGAAQA	None reported	[106]
*Palmaria palmata*	Enzyme hydrolysis with thermolysin, pH 8, 3 h,70 °C	ACE-1 inhibition	LDY, LDY, LRY, FEQWAS	0.14 µM 6.1 µM 0.044 µM 2.8 µM	[107]
*Porphyra yezoensis*	Enzyme hydrolysis with alcalase^®^	ACE-1 inhibition	n/a	1.6 g/L	[108]
*Porphyra yezoensis*	Enzyme hydrolysis with alcalase^®^	ACE-1 inhibition	n/a	0.516 g/L	[109]
*Palmaria palmata*	Osmotic shock/ precipitation with ammonium sulphate and papain hydrolysis	Renin inhibition	IRLIIVLMPILMA	3.344 mM	[110]
*Palmaria palmata*	Enzyme hydrolysis with papain	PAF-AH inhibition	NIGK	2.32 mM	[111]
*Porphyra spp.*	Enzyme hydrolysis with pepsin (E:S 8), pH 2, 4 h, 37 °C and viscozyme^®^	α-amylase inhibition	GGSK ELS	2.58 mM 2.62 mM	[112]
*Porphyra yezoensis*	Chemically synthesised and purified by HPLC	COX-2 inhibition	PPY1	n/a	[113]
**Phaeophyta (Brown)**					
**Seaweed Species**	**Extraction Process**	**Reported Bioactivity**	**Amino Acid Sequence**	**IC_50_**	**Reference**
*Undaria pinnatifida*	Enzyme hydrolysis with protease S “amano” pH 8, 18 h, 70 °C	ACE-1 Inhibition	VY, IY, AW, FY, VW, IW, LW	35.2 μM, 6.1 μM, 18.8 μM, 42.3 μM, 3.3 μM, 1.5 μM, 23.6 μM	[114]
*Hizikia fusiformis*	Enzyme hydrolysis with pepsin, pH 2,	ACE-1 Inhibition	SKTY, GKY, SVY	8.12μM, 3.92μM, 11.07μM	[115]
*Saccharina longicruris*	Enzyme hydrolysis with trypsin (E:S 0.05), pH 7, 24 h, 30 °C	Anti-bacterial activity	TITLDVEPSDTIDGVK, ISGLIYEER, MALSSLPR, ILVLQSNR, ISAILPSR, IGNGGELR, LPDAALN, EAESSLTGGNGCAK, QVHPDTGISK	None reported	[116]
*Laminaria japonica*	Enzyme hydrolysis with alcalase^®^ (60°C, pH 8) papain (50 °C and pH 7), and trypsin (pH 7.5, 55 °C)	ACE inhibition	KY, GKY, STKY, AKY, AKYSY, KKFY, FY, KFKY	5.24 μM, 7.94 μM, 20.63 μM, 7.52 μM, 2.42 μM, 15.33 μM, 4.83 μM, 10.73 μM	[117]
*Spirulina platensis*	Ultrasound and subcritical water	α-amylase inhibition	LRSELAAWSR	313.6 μg/mL	[118]
*Spirulina platensis*	Ultrasound and subcritical water	α-glucosidase	LRSELAAWSR	134.2 μg/mL	[118]
*Sargassium maclurei*	Enzyme hydrolysis with pepsin and papain	ACE-1 inhibition	RWDISQPY	72.24 μM	[119]

**Table 4 foods-11-00571-t004:** Uses of different seaweed in human food.

Seaweed Species	Product	Function	Reference
*Caulerpa racemosa*	Semi sweet biscuits	Functional antioxidant	[65]
*Ulva intestinalis*	Fish surimi	Functional and antioxidant effects	[120]
*Ulva lactuca*	Pork patties	Natural antioxidants	[121]
*Ulva rigida*	Pork patties	Natural antioxidants	[121]
*Himanthalia elongata*	Meat based products	Antioxidant	[122]
*Himanthalia elongata*	Breadsticks	Functional ingredient	[122]
*Fucus vesiculosus*	White bread	Antioxidant	[123]
*Fucus vesiculosus*	Pasta	Functional ingredient	[124]
*Undaria pinnatifida*	Milk	Functional ingredient	[125]
*Laminaria* sp.	Fish feed	Functional ingredient	[126]
*Phyllophara* sp.	Animal feed	Functional ingredient	[127]
*Porphyra yezoensis*	Fish feed	Functional ingredient	[128]
*Grateloupia turuturu*	Food colouring	Functional ingredient	[129]

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
