# Peer review of "Macroalgal Proteins: A Review"

_foods, 2022, doi:10.3390/foods11040571_

Round 1

Reviewer 1 Report

The authors have presented the review article 'Macroalgal proteins' in an appropriate way. In this manuscript, they have explained various methods of protein extraction taken seaweeds are explored especially in human foods along peptides. Flow charts are informative.

1.3.2.4. Application of red seaweeds in foods and feeds

In this part, feed portion is missing. 

Apart, the authors have provided all the information which are previously published but how they connect it all is the question. Did they interconnected all?

Author Response

Reviewer comments 1: RESPONSE

The authors have presented the review article 'Macroalgal proteins' in an appropriate way. In this manuscript, they have explained various methods of protein extraction taken seaweeds are explored especially in human foods along peptides. Flow charts are informative.

Response: We thank the reviewer for their comments especially regarding flowcharts.

1.3.2.4. Application of red seaweeds in foods and feeds

In this part, feed portion is missing.

Apart, the authors have provided all the information, which are previously published, but how they connect it all is the question. Did they interconnected all?

Response: We thank the reviewer for their comments. We have included the following text and references in relation to this query and hope that this satisfies the reviewers concerns:

Red seaweeds also have applications as feed ingredients for animals and fish/aquaculture. In recent times, they are used for their anti-methane effects when used as additives in the diet of ruminants including cattle, sheep and dairy cows. Several authors have reported that the red seaweed Asparagopsis taxiformis when added to feeds at concentrations between 0.5 to 4% of the total dry matter intake can reduce methane emissions by as much as 98% [142, 143]. However, the active agent in this instance is the bioactive compound bromoform – which is a known carcinogen. Red seaweeds can also be used in animal feeds as a source of anthelmintic agents. Recently, the anthelmintic action of a traditional remedy developed in Italy consisting of the red seaweeds Palisada tenerrima, Laurencia intricata and Laurencia spp. red algae was assessed using the egg hatch test [144]. An egg hatch reduction of 89.5, 43.7, and 23.4% was observed at 50, 5 and 1% dilutions [144].”

Reviewer 2 Report

The paper “Macroalgal proteins: A review” is a comprehensive collection about the properties of seaweed and pectin extraction. I really appreciate the authors for the extensive collection of literature and the nice presentation. However, the following specific points should be considered.

Abstract

It is looking very general. Please add the significant findings and mechanisms

Keywords: Avoid the words used in the title

Introduction

L36: Please highlight the anti-nutritional factor present in the seaweed

Highlight the different applications of seaweed protein in the food industry

Write the novelty of this study before objectives

L94: Introduce a new section “Search Methodology” to explain about the search engines used, number of articles collected, criteria used to shortlist the articles, etc.,

Add a schematic diagram to explain the principle of different extraction method

L122-132: Explain about the effect of microwave processing parameters on the extraction efficiency

Before reaching the conclusion section add a new section on “Industrial Scale technologies for Protein Extraction from Seaweed”

Also, add a new section on “Challenges associated with extraction technologies and future research prospects”

Reference

The old references (Published before 2010) should be replaced with recent references.

Author Response

Response to reviewer comments 2

The paper “Macroalgal proteins: A review” is a comprehensive collection about the properties of seaweed and pectin extraction. I really appreciate the authors for the extensive collection of literature and the nice presentation. However, the following specific points should be considered.

Response to reviewer comments: We would like to take this opportunity to thank the reviewer for their comments, which we aim to address in this response to improve the review paper.

Abstract

It is looking very general. Please add the significant findings and mechanisms

Response: We have added the following sentence to reflect the significant findings and mechanisms used to collate the review. Please see: “

“The paper discusses the significant uses of  seaweeds which range from use as anthelmintic and anti-methane ingredients used as feed additives to food techno-functional ingredients in the formulation of human foods such as ice-creams to health beneficial ingredients to reduce high blood pressure and prevent inflammation. This information was collated following a review of 206 publications on the use of seaweeds as foods and feeds and processing methods to extract seaweed proteins.”

Keywords: Avoid the words used in the title

Response: We have removed the words macroalgae and protein from the keywords as they are listed in the title (as suggested by the reviewer).

Introduction

L36: Please highlight the anti-nutritional factor present in the seaweed

Response: We have highlighted the anti-nutritional factors present in the seaweed as follows:

However, seaweeds also can contain bioactive compounds that may have anti-nutritional effects if consumed in high concentrations (usually above 4% dry matter intake for animal feeds) including phlorotannins, polyphenols and lectins as well as saponins and flavanoids. These also have bioactivities and health effects associated with their consumption in suitable concentrations.”

Highlight the different applications of seaweed protein in the food industry

Response: We have included the following text to highlight the different applications of seaweed proteins in the food industry: “In the food industry, seaweeds are consumed whole and dried, in pasta and bread products, as salads and as supplements currently. Extracts from seaweeds are used a weight control agents and as antioxidants and are used to make agar for microbiology purposes as well as food ingredients due to their technofunctional attributes. Macroalgal proteins are used in the manufacture of alternatives to meat patties and burgers and macroalgal proteins also find use in egg substitute products including mayonnaise and eggs.”

 Write the novelty of this study before objectives

Response: We included the following text: “The novelty of this review is that it discusses macroalgal proteins for use in foods and feeds.”

L94: Introduce a new section “Search Methodology” to explain about the search engines used, number of articles collected, criteria used to shortlist the articles, etc.,

Response: We have included a search methodology section in the introduction and detail the number or articles collated and criteria used to shortlist articles. “The novelty of this review is that it discusses macroalgal proteins for use in foods and feeds. This review collates the existing detailed information from previously published literature on methods for extraction of proteins and bioactive peptides from red, green and brown seaweeds using enzymes and potential applications of extracted proteins and peptides in an objective review of over 206 papers with original papers cited”

Add a schematic diagram to explain the principle of different extraction method

Response: We have added Table 1 which, explains the principles of different extraction methods sufficiently

L122-132: Explain about the effect of microwave processing parameters on the extraction efficiency

Response: Microwave processing is explained in the following: “The extraction of target compounds occurs through the application of electromagnetic radiation resulting in the breakdown of the bonds within the algal cell wall. The main advantage is that microwave assisted extraction (MAE) rapidly heats the sample solvent mixture and results in wide applicability for the rapid extraction of analytes, including thermally unstable substances ”.

Before reaching the conclusion section add a new section on “Industrial Scale technologies for Protein Extraction from Seaweed”

Response: We have added the following section to the paper as requested

 “2.1 Industrial scale technologies for the extraction of protein from seaweeds

Ultrasound and MAE are available as protein disruption methodologies at industry scale as are methods including high pressure process (HPP). Membrane filtration technologies including ultrafiltration and diafiltration methods may be applied to seaweeds such as Grateloupia turuturu to extract phycobiliproteins known as R-phycoerythrin at pilot plant scale. In this process, industrial type polyethersulfone 25–30 kDa membranes are used [319]. Results indicate that 100% of R-phycoerythrin can be recovered along with 32.9% of other proteins found in the seaweed and antinutritional compounds. Labour is the main expense of this process as well as maintaining the integrity of the membranes [319]. If this process could be adopted for extraction and isolation of seaweed proteins and bioactive peptides could be rapidly increased, greatly enhancing their potential use in pharmaceutical, cosmetic and industrial applications.”

Also, add a new section on “Challenges associated with extraction technologies and future research prospects”

Response: We have added the following section under the Industrial section of the paper: “ Challenges associated with extraction technologies include costs including labour costs and cleaning costs as well as environmental hazards associated with chemicals used for membrane cleaning processes. Other challenges include sensory and taste challenges associated with end macroalgal protein products – i.e. colour can limit the application of some macroalgal proteins and hydrolysates”.

Reviewer 3 Report

For Authors of the manuscript ID foods-1547800 Title: "Macroalgae proteins: A review".
In my opinion, this manuscript could be considered as such.
I justified my decision because the review of macroalgae proteins is entirely devoted to the main topic.
The literature review is based on many publications (321 items), mainly from the last ten years. The authors thoroughly described the ingredients of seaweed and how to obtain them. The data presented in the tables and graphs are based on the analysis of many research works.
The authors emphasize that in the present time of protein deficit, if necessary, seaweed can be used as a source of protein and bioactive peptides, that can be used to promote human and animal health. Therefore, seaweed can be used on a larger industrial scale in the agriculture, food and pharmaceutical industries. Additionally, the authors reported that enzyme assisted extraction could be successfully used to increase the yield of extracted protein to generate bioactive peptides with health properties. However, more work is needed to improve products and test other seaweed species for bioactivity, including the activities of various enzymes (renin, cyclooxygenase, α-amylase and α-glucosidase), mainly in vivo.

Author Response

Response to reviewer comments 3

For Authors of the manuscript ID foods-1547800 Title: "Macroalgae proteins: A review".

In my opinion, this manuscript could be considered as such.

Response: We thank the reviewer for their comments.

I justified my decision because the review of macroalgae proteins is entirely devoted to the main topic.

The literature review is based on many publications (321 items), mainly from the last ten years. The authors thoroughly described the ingredients of seaweed and how to obtain them. The data presented in the tables and graphs are based on the analysis of many research works.

The authors emphasize that in the present time of protein deficit, if necessary, seaweed can be used as a source of protein and bioactive peptides, that can be used to promote human and animal health. Therefore, seaweed can be used on a larger industrial scale in the agriculture, food and pharmaceutical industries. Additionally, the authors reported that enzyme assisted extraction could be successfully used to increase the yield of extracted protein to generate bioactive peptides with health properties.

Response: We thank the reviewer for their synopsis of the paper.

However, more work is needed to improve products and test other seaweed species for bioactivity, including the activities of various enzymes (renin, cyclooxygenase, α-amylase and α-glucosidase), mainly in vivo.

Response: We agree with the reviewer and we have added more information on the enzymes described – alpha amylase and alpha glucosidase and we also mention in the revised manuscript that it is necessary to do further testing in vivo to prove the bioactivities associated with peptides of seaweed origin or derived from seaweeds.

We have included the following text:

Moreover, enzyme assisted hydrolysis has a number of advantages – hydrolysis is reported to increase protein yield and the isolation of bioactive peptides with associated health benefits. Further, it is though to reduce the potential of seaweed proteins to cause allergy. However, enzyme processes can be expensive [54, 55, 56, 40, 41]. In relation to the reported bioactivities associated with seaweed derived peptides, it is important that these bioactivities are confirmed in  models and animal trials. This may also pose ethical concerns around the use of animals in food product testing. However, until in vitro models more closely mimic animal models, this is a necessary step if producing seaweed proteins and hydrolysates for potential health products where in order to make a claim, the product must be proven to comply with existing legislation in Europe or America governed by the European Food Safety Authority (EFSA) or the FDA, respectively.

Round 2

Reviewer 2 Report

Great work on revising the manuscript. I am recommending acceptance of the revised manuscript.